# Optimising broadband pulses for DEER depends on concentration and distance range of interest

Andreas Scherer, Sonja Tischlik, Sabrina Weickert, Valentin Wittmann, Malte Drescher

Department of Chemistry and Konstanz Research School Chemical Biology, University of Konstanz, Konstanz, Germany

*Correspondence to:* Malte Drescher (malte.drescher@uni–konstanz.de)

**Abstract.** EPR distance determination in the nanometre region has become an important tool for studying the structure and interaction of macromolecules. Arbitrary waveform generators (AWGs), which have recently become commercially available for EPR spectrometers, have the potential to increase the sensitivity of the most common technique double electron-electron resonance (DEER, also called PELDOR), as they allow the generation of broadband pulses. There are several families of

broadband pulses, which are different in general pulse shape and the parameters that define them. Here, we compare the most common broadband pulses. When broadband pulses lead to a larger modulation depth they also increase the background decay of the DEER trace. Depending on the dipolar evolution time this can significantly increase the noise level towards the end of the form factor and limit the potential increase of the modulation-to-noise ratio (MNR). We found asymmetric hyperbolic secant (HS{1,6}) pulses to perform best for short DEER traces leading to a MNR improvement of up to 86 % compared to

rectangular pulses. For longer traces we found symmetric hyperbolic secant (HS{1,1}) pulses to perform best, however, the increase compared to rectangular pulses goes down to 43 %.

## 1 Introduction

In the last years DEER (double electron-electron resonance) has developed into an important technique for the determination of distances in the nanometre range (Jeschke, 2012, p.2; Milov et al., 1984; Salkhon, K.M. Milov, A.D., Shchirov, M.D., 1981)

and in particular into a suitable tool for studying biological macromolecules (e.g. proteins (Jeschke, 2012; Robotta Marta et al., 2014) or RNA/DNA (Grytz et al., 2017; Kuzhelev et al., 2018)). As many bio-macromolecules do not contain paramagnetic centres, for many DEER experiments spin labels are introduced with the help of site-directed spin labelling (Hubbell et al., 1998). Although many different types of spin labels have been introduced in the last years ranging from trityl (Abdullin et al., 2015; Jassoy et al., 2018), Gd(III) (Collauto et al., 2016; Dalaloyan et al., 2015; Mahawaththa et al., 2018), Copper(II) (Wort

et al., 2019) to photoexcitable spin labels (Di Valentin et al., 2014; Hintze et al., 2016), just to mention a few examples, nitroxide labels are still amongst the most widely used tags.

Increasing the sensitivity of DEER spectroscopy is an active field of research (Borbat et al., 2013; Breitgoff et al., 2017; Doll et al., 2015; Jeschke et al., 2004; Lovett et al., 2012; Milikisiyants et al., 2019; Polyhach et al., 2012; Tait and Stoll, 2016; Teucher and Bordignon, 2018). A very elegant approach to increasing DEER sensitivity has been made possible by the

availability of arbitrary waveform generators with time resolution in the nanosecond region as they allow the generation of broadband microwave pulses (Doll et al., 2013; Doll and Jeschke, 2017; Spindler et al., 2017).

Here, we compare nitroxide-nitroxide DEER performance for different types of pulses and different sample conditions. The manuscript is organized as follows: In Sect. 1 we will give a brief overview over the pulse shapes that are compared in this manuscript. In Sect. 2, we will describe the experimental details and the compounds that have been used in this study. In Sect.

3, we will present and discuss the experimental results. We will compare rectangular, Gaussian and different  types of broadband shaped pulses on a commercial spectrometer. In order to give them a fair comparison, the parameters for each pulse family will be optimised. In doing so, we will provide a step-by-step guidance how an experimental optimisation for DEER

can be performed. The larger inversion efficiency of broadband shaped pump pulses that leads to a higher modulation depth will also lead to a higher background decay and therefore potentially limit the signal gain that is promised by broadband shaped pump pulses. We set out to examine this effect for the presented pulse families in detail and show that different types of broadband shaped pulses are ideal for different spin concentrations and distance ranges.

In magnetic resonance experiments, a pulse is generated by a time-dependent field $B_1$ that is applied perpendicular to the $B_0$ field which defines the z-direction. All pulses in this paper can be described in terms of an amplitude function $A(t)$ and a frequency function $\omega(t)$.

The resulting $B_1$ field in the rotating frame is:

$$B_{1,x}(t) = A(t)\cos\big(\rho(t)\big), \tag{1}$$

$$B_{1,y}(t) = A(t)\sin\big(\rho(t)\big). \tag{2}$$

Where the phase $\rho(t)$ is defined as $\rho(t) = \int_0^t \omega(t')dt'$. Rectangular pulses are described by $\omega(t) = 0$ and $A(t) = B_1$ during the pulse, i.e. by a $B_1$ field with a constant phase and intensity. The sidebands of the sinc-shaped excitation profile of rectangular pulses increase the overlap of the observer and pump pulse in DEER, resulting in so called '2+1' artefacts at the end of the DEER trace. It has recently been shown that those artefacts can be reduced by replacing the rectangular pulses with

Gaussian pulses (Teucher and Bordignon, 2018). Gaussian pulses also have a frequency function of $\omega(t) = 0$ but an amplitude function:

$$A(t) = \exp\left(-\frac{4\ln(2)t^2}{\text{FWHM}^2}\right), \tag{3}$$

FWHM describes the full width at half maximum of the pulse in the time domain (Teucher and Bordignon, 2018). Here and in the following equations the time axis is defined such that $t = 0$ lies in the centre of the pulses. During a rectangular or

Gaussian pulse the magnetisation vector is rotated around the $B_1$ field with an angle that is independent of the initial orientation of the magnetisation vector. Such pulses are therefore called uniform rotation pulses (Kobzar et al., 2012). As rectangular and Gaussian pulses have a fixed frequency, they are also referred to as monochromatic pulses.

One of the most significant challenges in EPR spectroscopy is the limited excitation bandwidth of rectangular and also Gaussian pulses compared to the width of many EPR spectra. In the case of nitroxide-nitroxide DEER, a significant part of the

EPR spectrum does neither contribute to observing nor to pumping when using rectangular pulses.

Using broadband shaped pulses, the excitation bandwidth can be increased (Doll et al., 2013). Broadband shaped pulses distinguish from rectangular and Gaussian pulses mainly in that they do not have a constant frequency, but the frequency is swept over a given range during the pulse, which allows increasing the excitation bandwidth. In an accelerated frame, which rotates with the instantaneous excitation frequency of the pulse, the effective field rotates from the +z to the –z direction (Baum

et al., 1985; Deschamps et al., 2008; Garwood and DelaBarre, 2001; Kupce and Freeman, 1996). Under adiabatic conditions the magnetisation follows the effective field on its way from +z to –z (Baum et al., 1985; Doll et al., 2013a). Pulses that induce this kind of spin flip behaviour are called point-to-point rotation pulses. This approach allows the generation of pulses that have a large excitation bandwidth and that are, above a certain threshold, more insensitive to the resonator profile than rectangular pulses (Baum et al., 1985). Their ability to flip spins from the +z to the –z-axis makes such broadband shaped

pulses perfect candidates for the pump pulse in the DEER pulse sequence. Their larger excitation profile has the potential to result in a larger modulation depth and therefore a larger sensitivity (Bahrenberg et al., 2017; Doll et al., 2015; Spindler Philipp E. et al., 2013; Tait and Stoll, 2016).

Intuitively, a high adiabaticity means that the effective magnetic field moves more slowly from +z to –z, making it easier for the spins to follow, thus resulting in a higher inversion efficiency.

The adiabaticity $Q$ is formally defined as (Kupce and Freeman, 1996):

$$Q = \frac{2\,\pi v_1}{|\mathrm{d}\theta/\mathrm{d}t|}. \tag{4}$$

Here, $v_1$ is the strength of the effective magnetic field and $\theta$ is its polar angle in the accelerated frame. The pulses have a good inversion efficiency, if $Q \gg 1$ (Deschamps et al., 2008). In general, the adiabaticity changes during the duration of the pulse and is different for spins with different frequency offsets. Adiabatic pulses are typically quantified by their minimum adiabaticity $Q_{\mathrm{min}}$.

Chirp pulses have a constant amplitude function and a linear frequency function $\omega(t) = f_{\mathrm{start}} + pt$, where $p = \frac{\Delta f}{t_p}$ is a sweep

constant, $t_p$ is the pulse length and $\Delta f = f_{\mathrm{end}} - f_{\mathrm{start}}$. $f_{\mathrm{start}}$ and $f_{\mathrm{end}}$ are the start and end frequencies of the frequency sweep. The minimum adiabaticity $Q_{\mathrm{min}}$ is reached when a spin is on resonance with the pulse frequency (Doll et al., 2013a):

$$Q_{\mathrm{min}} = \frac{2\pi v_1^2 t_p}{\Delta f}, \tag{5}$$

$Q_{\mathrm{min}}$ increases with the pulse length but decreases with the sweep width. The frequency width for a pump pulse should be chosen such that a large part of the spectrum is excited without having significant spectral overlap with the pulses at the

observer frequency. The steep flanks at the beginning and the end of the rectangular amplitude profile lead to distortions in the excitation profiles of chirp pulses, because the initial effective magnetic field is not aligned with the z-axis. Smoothing both ends of the pulses with a quarter sine-wave can reduce theses distortions (Bohlen and Bodenhausen, 1993). The smoothing can be adapted by changing the rising time $t_{\mathrm{rise}}$. Following the logic so far, the pulse length should be chosen as long as possible to enable a very high adiabaticity. However, a broadband shaped pulse flips spins with different offsets at different

times. When used as a pump pulse in DEER, this results in a shift of the dipolar oscillations and an artificial broadening of smaller distances in the distance distribution. Therefore, the pulse length should be chosen such that (Breitgoff et al., 2019):

$$t_p < \frac{T_{dd}}{4}, \tag{6}$$

with the dipolar evolution time $T_{dd}$ of the shortest expected distance.

In addition to chirp pulses there are more elaborate pulses employing more elaborate frequency and amplitude functions. The

most common ones are WURST (wideband, uniform, smooth truncation) and HS (hyperbolic secant) pulses. The trends discussed so far are valid for them as well. However, they feature additional parameters that can be used to tune the steepness of the corresponding excitation profiles.

WURST pulses have a linear frequency sweep as well but a different amplitude function than chirp pulses (Kupče and Freeman, 1995b; Spindler et al., 2017):

$$A(t) = A_{\mathrm{max}}\left(1 - \left|\sin\left(\frac{\pi t}{t_p}\right)\right|^n\right), \tag{7}$$

The effect of the parameter $n$ determining the steepness of the amplitude function will be discussed below.

HS pulses have non-linear frequency sweeps and are described by the following amplitude and frequency functions:

$$A(t) = \mathrm{sech}\left(\beta 2^{h-1}\left(\frac{t}{t_p}\right)^h\right), \tag{8}$$

$$\omega(t) = \frac{\Delta f}{2} \tanh\left(\frac{\beta}{2}\right)^{-1} \tanh\left(\frac{\beta t}{t_p}\right), \tag{9}$$

with order parameter $h$ and truncation parameter $\beta$. The effects of $\beta$ will be discussed below. A common choice for $h$ is to set $h = 1$. These pulses have an offset-independent adiabaticity and a rather rectangular excitation profile (Baum et al., 1985; Tannús and Garwood, 1996). Increasing the order $h$ of an HS pulse will lead to a higher adiabaticity at the maximum of the excitation profile but less steeper flanks (Breitgoff et al., 2019). A compromise can be found by using an asymmetric HS pulse where the flank close to the observer is made steep by an order of 1 and where the other flank has a higher order for a higher adiabaticity (Doll et al., 2016). Symmetric pulses with an order parameter of $h = 1$ will be referred to as HS{1,1}, asymmetric pulses where the first part of the pulse has on order parameter of $h = 1$ and the second half has $h = 6$, as suggested by Doll et al. (2016) are referred to as HS{1,6} (Doll et al., 2016).

The measured DEER-trace $V(t)$ is the product of the form factor $F(t)$ that contains the required intramolecular distance information and a background-function $B(t)$ (Jeschke, 2012):

$$V(t) = F(t) \cdot B(t), \tag{10}$$

The background decay is caused by the intermolecular interactions of the observer spin with pump spins of surrounding molecules. Assuming that the spins are homogenously distributed the background decay can be described by an exponential decay (Jeschke, 2007a):

$$B(t) = \exp(-(k|t|)^{d/3}), \tag{11}$$

where d is a dimensionality constant and the decay constant $k$ is described by the following equation (Hu and Hartmann, 1974; Pannier et al., 2000):

$$k = \frac{2N_a \pi \mu_0}{9\sqrt{3}\hbar} g^2 \mu_e^2 f c, \tag{12}$$

Here, $c$ is the spin concentration, $f$ the inversion efficiency of the pump pulse, $\mu_e$ the Bohr-magneton, $\mu_0$ the magnetic field constant, $N_a$ the Avogadro number and $g$ the isotropic g-factor of the nitroxide.

## 2 Materials and Methods

### 2.1 Sample preparation

Wheat germ agglutinin (WGA) was purchased from Sigma-Aldrich (article-no.: L9640) as lyophilized powder and used without further purification. The doubly spin-labelled tetravalent ligand (1) was synthesised in the lab of Valentin Wittmann. Details of synthesis and characterisation will be published elsewhere. For the WGA-ligand samples investigated in this study solutions of WGA and the tetravalent ligand were prepared separately in deionised water. The protein concentration of the WGA solution was determined spectrophotometrically.

WGA-ligand samples were prepared by mixing WGA and ligand solutions resulting in a 2:1 molar excess of WGA compared to the ligand referring to the final sample volume. The 2-fold excess on protein was chosen to prevent free, unbound ligand in solution. The sample solution was lyophilised and the resulting powder was dissolved in D$_2$O (Magnisolv, Cas-no.: 7789200, article: S571556621) and 20 % (v/v) deuterated glycerin (Sigma-Aldrich, lot-no. MBBB5255, article: 447498-1G) as cryoprotectant. Unless stated otherwise we used a sample concentration of 160 μM WGA and 80 μM ligand. 60 μL of solution were filled into 3 mm outer diameter quartz sample tubes (ER 221 TUB/2, Part No. E221003), shock-frozen in liquid nitrogen before measurement and placed in the probe head precooled to 50 K. Samples were stored at -80 °C with unfreezing avoided.

**2.2 EPR experiments**

All experiments have been performed on a Bruker Elexsys E580 spectrometer at Q-band (34 GHz). The spectrometer is equipped with a SpinJet-AWG unit (Bruker) and a 150 W pulsed travelling-wave tube (TWT). All samples were measured in 3 mm outer diameter sample tubes in an overcoupled ER5106QT-2 resonator (Bruker). The quality factor Q of the overcoupled
resonator is approximately 200.

The samples were cooled to 50 K with a Flexline helium recirculation system (CE-FLEX-4K-0110, Bruker Biospin, ColdEdge Technologies) comprising a cold head (expander, SRDK-408D2) and a F-70H compressor (both SHI cryogenics, Tokyo, Japan), controlled by an Oxford Instruments Mercury ITC.

DEER measurements were recorded with the standard four pulse DEER sequence (Pannier et al., 2000), an 8-step phase cycle
(Tait and Stoll, 2016) and nuclear modulation averaging (Jeschke, 2012). The dipolar evolution time was set to 8 μs and the time step to 8 ns.

We analysed the DEER traces with DeerAnalysis2019 (Jeschke et al., 2006). We performed a background correction resulting in a background function with a dimension of $d = 3.5$. The form factor was analysed with Tikhonov regularisation and a regularisation parameter chosen by the generalised cross-validation criterion (Edwards and Stoll, 2018).

A crucial parameter for pulsed dipolar spectroscopy is the modulation-to-noise parameter $\text{MNR} = \frac{\lambda}{n}$, with the modulation depth $\lambda$ and the noise level $n$. We calculated the noise similarly to published procedures by the standard error from a fit with a smoothing spline (Bahrenberg et al., 2017; Breitgoff et al., 2019; Mentink-Vigier et al., 2013). We excluded the first 10 datapoints from the form factor because the spline typically showed some deviations at the start of the trace. Unless stated otherwise the upper limit for the noise calculation was 7 μs.

We used the $\eta_{2P}$-parameter which has been suggested by (Doll et al., 2015) and already been used by other authors (Doll et al., 2015; Spindler Philipp E. et al., 2013; Tait and Stoll, 2016). The $\eta_{2p}$ value is defined as the difference between two distinct time points in the DEER trace, and therefore does not require the measurement of full DEER traces. We recorded short DEER traces with 8 data points only and calculated $\eta_{2P}$ as the difference of the phase corrected DEER trace at the zero time $V(0)$ minus the first minimum of the DEER trace $V(t_{\min})$.

For a more detailed description of materials and methods see S1.

**3 Results and Discussion**

In order to study the performance of DEER using different pulses, we used the doubly nitroxide-labelled tetravalent ligand bound to wheat germ agglutinin dimer (WGA) as a model system (Fig. 1). The ligand binds with a very high affinity to WGA and features a narrow distance distribution (FWHM = 0.2 nm) at 5.1 nm (to be published elsewhere). We performed DEER
experiments with different combinations of pulses. In the following, we will refer to a combination of rectangular observer and pump pulses as RR, to a combination of Gaussian observer and pump pulses as GG for, to a combination of rectangular observer and broadband shaped pump pulses as RS and to a combination of Gaussian observer and broadband shaped pump pulses as GS for.

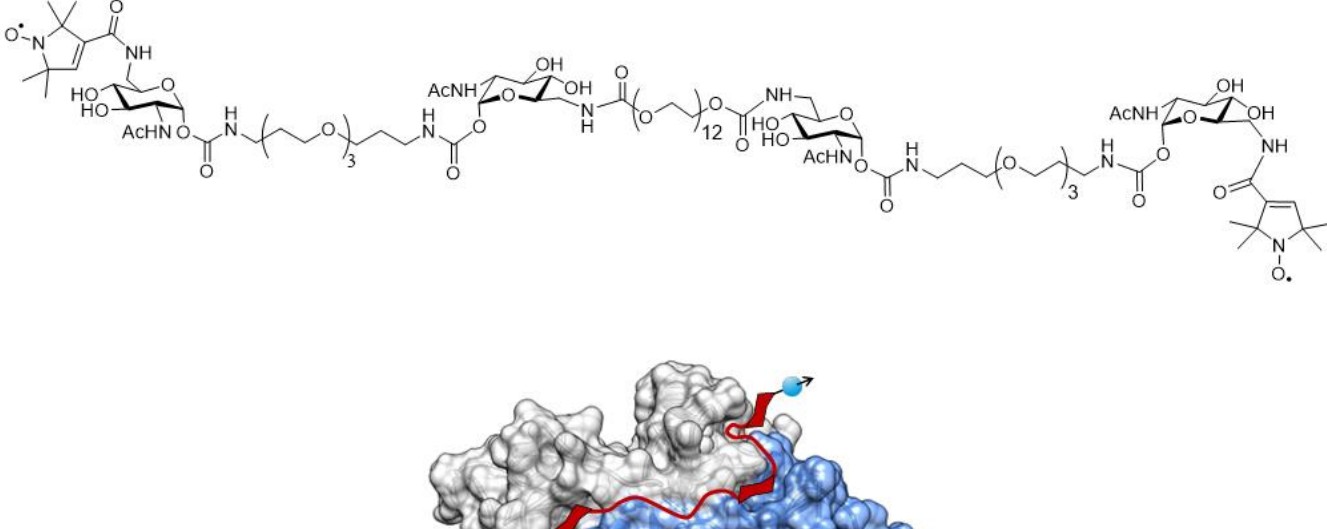

**Figure 1:** The structure of the tetravalent ligand with its two spin-(2,2,5,5-tetramethyl-3-pyrrolin-1-yloxycarbonyl)-labels and the ligand bound to WGA. The visualisation of the dimeric WGA with the subunits coloured in grey and blue is based on the crystal structure (PDB entry 2X52 (Schwefel et al., 2010)). In red, a schematic representation of the ligand is overlaid to the crystal structure. The ligand is suggested to bind with its four GlcNAc moieties to the primary binding sites of WGA. Blue balls and arrows indicate nitroxide spin labels.

As stated by equation (10) the measured raw data does not only consist of the desired form factor but includes a background contribution emerging from intermolecular interactions. A common way to deal with this, is to fit the background according to equation (11) and divide the raw data by the fit to obtain the form factor that can then be transformed into a distance distribution (Jeschke, 2012; Jeschke et al., 2006). When measuring DEER traces, a precise distance determination is desired. Since for an experimental parameter optimisation, the true underlying distance distribution is unknown, a metric is needed that is based on the recorded data. The MNR of the form factor is a suitable for this case as it increases with an increasing modulation depth and an increasing echo intensity. As the noise of the form factor increases towards its end due to the division by the background, the MNR goes down with a stronger background decay. It can therefore capture the fact that a larger background decay leads to less reliable distance distributions as has recently be investigated by (Fábregas Ibáñez and Jeschke, 2020) in a detailed study. In their paper they also suggest a different method for background correction that treats the background by directly including it in the kernel that is needed to calculate the distance distribution from the DEER trace. As this methods renders the calculation of a form factor redundant, a MNR cannot be directly obtained by it. Even though this new method has shown itself to give more reliable distance distributions in the case of large background decays its performance still drops with an increasing background. Therefore, we consider the MNR that is obtained by the background correction by division still as the best measure to optimise settings for a DEER measurements experimentally.

The evaluation of the noise of the entire DEER trace is not always feasible. It depends on the maximum distance $r_{\mathrm{max}}$ that shall be detected up to which part the form factor is of interest. Here, we truncated the form factor for the calculation of the MNR at three times of the oscillation period of the maximum distance that is of interest (Edwards and Stoll, 2018):

$$\tau_{\mathrm{truncation}} = 3 \frac{\left(\frac{r_{\mathrm{max}}}{\mathrm{nm}}\right)^3}{52\ \mathrm{MHz}}. \tag{13}$$

This corresponds to roughly three dipolar oscillations in the form factor. In this case of a distance at 5.1 nm this is equivalent to a truncation time $\tau_{\text{truncation}} \approx 7\ \mu s$. A simulation with a model distance reveals that in order to obtain the correct width of the distance distribution, a good MNR up to this time point can be necessary and a good MNR of only the first part of the form factor is not as reliable when the credibility of the obtained distance distribution is to be estimated. The details of this study can be found in S2.

### 3.1 Performance comparison for rectangular and Gaussian pulses

The pump pulse frequency was set to 34.00 GHz which is the maximum of the resonator profile (Fig. 2a). The magnetic field was set such that we pumped on the maximum of the nitroxide spectrum (Fig. 2b). To optimise the settings for RR and GG we tested observer pulses with a frequency offset of 90 MHz and 70 MHz between the pump and observer pulse, respectively. To check for different excitation profile widths of the observer pulse, we tested settings with an observer pulse amplitude of 100 % and 60 %. The pulse length was always adjusted to get $\pi/2$ and $\pi$ pulses. The observer pulse lengths for both tested frequency offsets were identical in all experiments owing to the similar values of the resonator profile at both observer frequencies (33.91 GHz and 33.93 GHz, Fig. 2a). For rectangular observer pulses the pulse lengths were 28 ns and 32 ns, for Gaussian observer pulses, they were 56 ns and 74 ns. For the pump pulse we kept the amplitude fixed at 100 %, which resulted in pulse lengths of 16 ns for rectangular and 34 ns for Gaussian pulses. As we used Gaussian pulses that were generated by Xepr, the FWHM of the Gaussian pulses was automatically defined by the software as $\text{FWHM} = \frac{t_p}{2\sqrt{2\ln(2)}}$ and we did not optimise this parameter. An overview over all observer pulse settings can be found in Table S1 and S2.

For optimum observation of the spin echo modulation in DEER traces, it has been suggested to record the echo in transient mode and then perform a digital integration over a product of the recorded echo with a Gaussian filter (Pribitzer et al., 2017). This procedure is not ideal for commercial spectrometers as the transient recording of the echo drastically increases the spectrometer overhead time. Therefore, we performed a direct integration of the spin echo. We optimised the integration window for each parameter set for a maximum MNR by recording a series of Hahn echoes. Compared with commonly used integration lengths equal to the $\pi$-pulse length for rectangular pulses (Jeschke, 2007b), we find settings where a 14 % increase in the SNR can be achieved by choosing a larger integration window. For Gaussian pulses, we find that it is typically preferable to choose integration windows that are shorter than the $\pi$-pulse length. More details can be found in S2.

**Table 1:** The rectangular and Gaussian pulses with the best performance.

| Pulse Type | Offset [MHz] | Obs. Amp. [%] | MNR | Mod depth $\lambda$ |
|------------|--------------|---------------|-----|---------------------|
| **RR** | 70 | 60 | 35 | 0.31 |
| **GG** | 70 | 100 | 41 | 0.31 |

The best MNR for the setting RR was found to be 35 (Table 1). It was achieved for an offset of 70 MHz and 60 % intensity. The best MNR for GG was 41 at an offset of 70 MHz as well and a pulse amplitude of 100 % (Table 1). This corresponds to a 17 % increase in the MNR of Gaussian pulses compared to rectangular pulses. This is in contrast to the findings of (Teucher and Bordignon, 2018), who found that Gaussian observer pulses have a slightly lower MNR that rectangular pulses. The exact reason for the deviating results is not entirely clear to us, we assume that this is due to their different setup with a homebuilt-resonator that that has slightly different properties as our commercial one.

As expected, the missing sidebands of the Gaussian pulses allow the usage of higher pulse amplitudes. This hints that for the chosen parameters the pulse overlap is indeed a limiting factor for rectangular pulses. The modulation depth for RR and GG is approximately 30 % in both cases, but Gaussian pulses seem to have the advantage of a higher echo intensity, probably due

to a lower pulse overlap. For RR and GG, the small offset of 70 MHz performed better than a larger offset of 90 MHz most likely due to the different echo intensities at the corresponding positions in the EPR spectrum (Fig. 2b). The results for all RR and GG setting can be found in the tables S3 and S4.

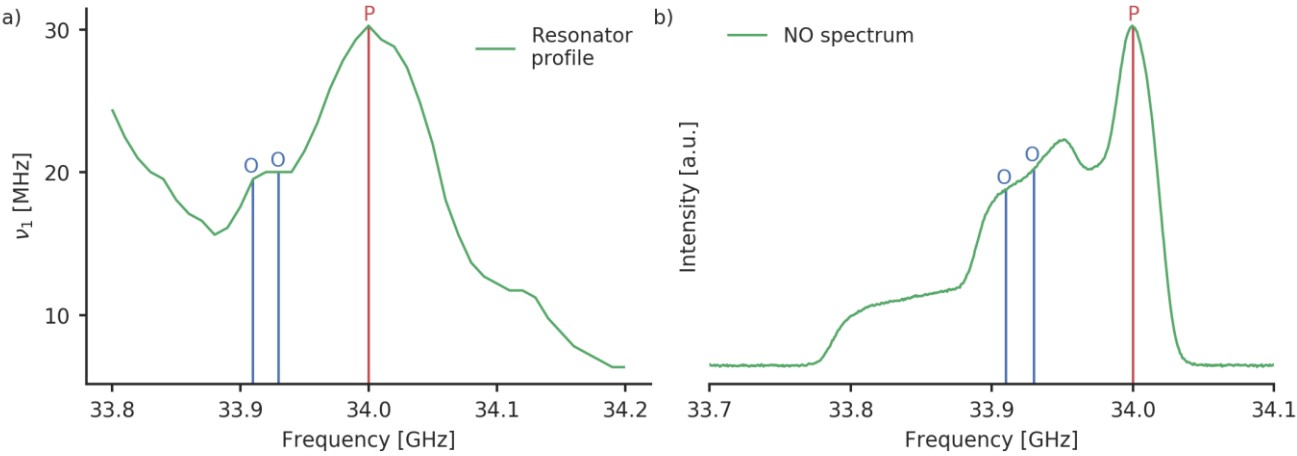

5    **Figure 2:** (a) Resonator profile with both tested observer frequencies (blue) and the  pump (red) frequencies for the rectangular pulses and (b) the nitroxide spectrum with the positions of the two tested observer frequencies and the pump frequency.

### 3.2 Broadband shaped pulses

We set out to investigate several broadband shaped pulses, i.e. chirp, WURST, HS{1,1} and HS{1,6} pulses for the settings RS and GS. Unless specified otherwise we used pump pulse lengths of 100 ns. According to Eq. (6), the pulse length of 100 ns
10   corresponds to a minimum accessible distance of $r_{min} = 2.75$ nm.  For the determination of shorter distances we also tested chirp pulses with a length of 36 ns (referred to as short chirp pulses below), which corresponds to a distance of $r_{min} = 1.96$ nm. Such a distance limit should be suitable for most practical applications.  Despite the fact that longer broadband shaped pump pulses should give higher inversion efficiencies, we found that they do not result in a better performance for DEER. As the minimum accessible distance also increases when longer pump pulses are used, we did not test pump pulses longer than 100 ns.
15   This is discussed in more detail in S13.

Spins are not flipped within the whole pulse duration but only a smaller fraction of it (Spindler Philipp E. et al., 2013). Simulations with an HS{1,1} pulse with a length of $t_p = 100$ ns, a frequency sweep width of $\Delta f = 110$ MHz and $\beta = 8/t_p$ show distances up to $r_{min} = 2.32$ nm could be detectable (see S7). It is, however, hard to generalise this effect as the spin trajectories for different broadband pulses are not necessarily the same.

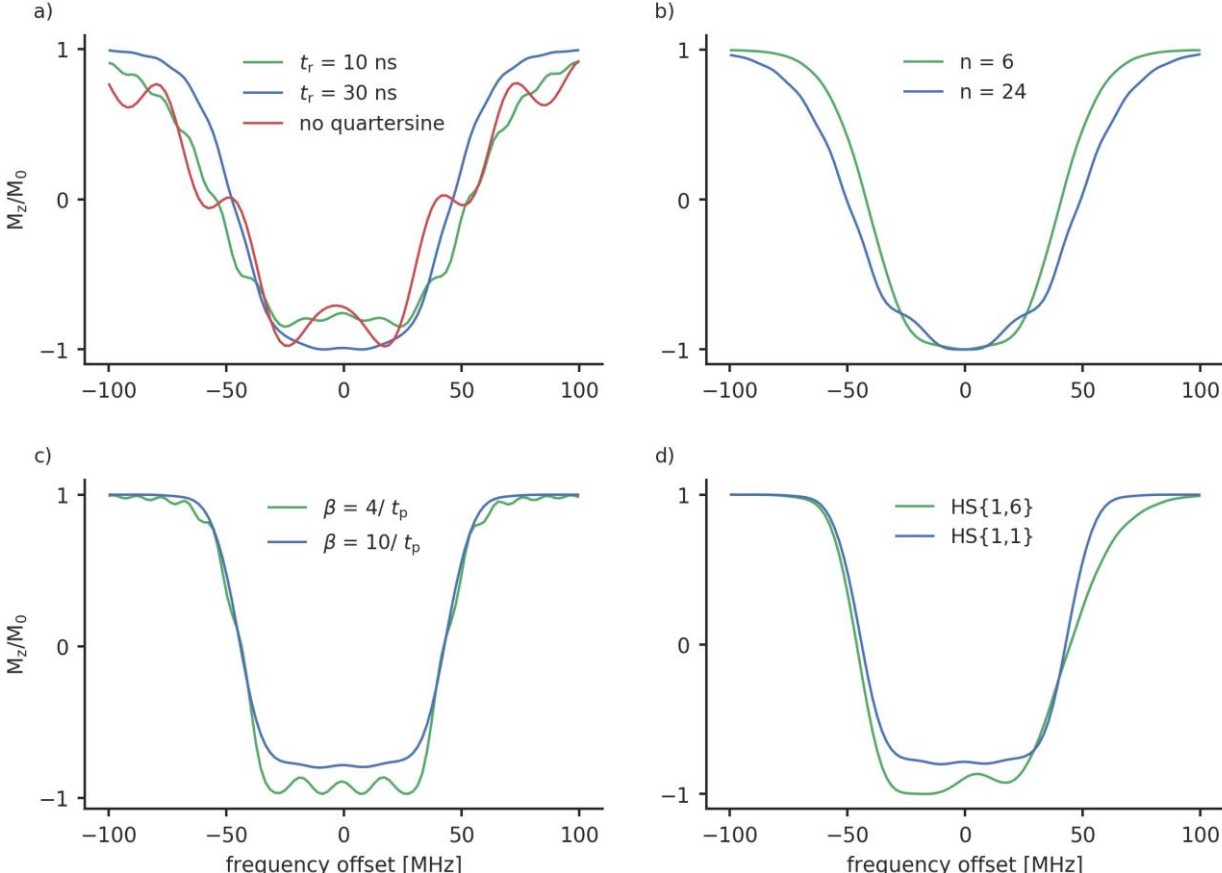

**Figure 3:** Calculated inversion profiles of broadband pulses normalised to $\nu_1$ = 30 MHz, which corresponds to the maximum of the measured resonator profile. a) Chirp pulses with a frequency width of 120 MHz, a length of 100 ns and a rising time of 10 ns (green), a rising time of 30 ns (blue), a length of 36 ns and no quarter sine smoothing (red). b) WURST pulses with a frequency width of 120 MHz, a pulse length of 100 ns and a value for n of 6 (green) and 24 (blue). c) HS{1,1} pulses with a frequency width of 90 MHz and a truncation parameter of 4 (green) and 10 (blue). d) An HS{1,6} (green) and HS{1,1} (blue) pulse with a width of 90 MHz and a pulse length of 100 ns. The truncation parameter was 10 in both cases.

Figure 3 shows the calculated excitation profiles of some of the tested pulses. The calculated excitation profiles are normalised to a $\nu_1$ field strength of 30 MHz, which we achieved with our setup at the maximum of the resonator profile. Under such conditions, the long chirp pulses have an adiabaticity of around 5, i.e. a chirp pulse with a length of 100 ns, a sweep width of 120 MHz and a $\nu_1$ strength of 30 MHz has a calculated adiabaticity of 4.7. A short chirp pulse with a length of 36 ns (and otherwise unchanged parameters) has an adiabaticity of 1.7 due to the higher frequency sweep rate. Although this value is rather low, the calculations show that short chirp pulses achieve a nearly complete inversion efficiency around the maximum of the excitation profile (Fig. 3a). On the other hand, the excitation profile is rather broad with many sidebands. The finite length of the pulses creates an additional distortion. By smoothing the edge with a quarter sine, this disturbance can be reduced (Fig. 3a). A higher rising time will lead to a more properly defined excitation profile with fewer sidebands but the overall width of the excitation profile is reduced (see Fig. 3a).

WURST pulses (Fig. 3b) are characterised by an additional parameter $n$. A high value of $n$ results in a more rectangular shape of the pulse and leads to distortions in the excitation profile around the maximum (Kupce and Freeman, 1995b, 1995a; O'Dell, 2013). Small values of $n$ lead to excitation profiles with very steep and well defined side flanks. However, for small $n$ very long pulse durations are needed to achieve a high inversion efficiency. As long pulses are not feasible, because they limit the

minimum distances that can be resolved, we chose to stick to 100 ns pulses and test the values for $n$ of 6, 12 and 24, for which a reasonable excitation profile can be expected (Fig. 3b).

In Fig. 3c we show the comparison of the excitation profile of HS{1,1} pulses for a truncation parameter of $\beta = 4/t_p$ and $\beta = 10/t_p$. For $\beta = 10/t_p$ the inversion efficiency is smaller than for $\beta = 4/t_p$, however, the excitation profile is well defined and does not show the sideband oscillations that can be seen for the latter.

Owing to their higher adiabaticity, HS{1,6} pulses feature higher inversion efficiency than HS{1,1} pulses with otherwise equal parameters (Fig. 3d), while maintaining the steep frequency flank towards the observer profile at the lower frequency end.

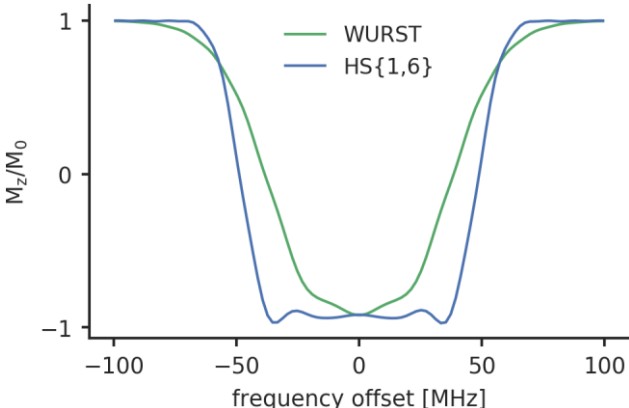

**Figure 4:** Calculated inversion profiles of a WURST ($n = 12$, green) and a HS{1,6} (truncation parameter of $6/t_p$, blue) pulse with a pulse length of 100 ns and a sweep width of 100 MHz normalised to $\nu_1 = 30$ MHz, which corresponds to the maximum of the measured resonator profile.

For HS{1,1} and HS{1,6} pulses a frequency sweep width of 50 MHz to 110 MHz were tested. WURST and chirp pulses tend to have a narrower excitation profiles for a given frequency sweep width at the tested parameters (see Fig 4). We therefore chose to use higher frequency sweep widths for WURST than for HS pulses to achieve a similar excitation bandwidth.

As the bandwidth of the resonator and the width of the spectrum is limited, there is an optimum offset between the two pulses that minimises the overlap but is not too large for the resonator bandwidth. We tested offsets from a range of 70 MHz to 130 MHz. The offset is defined as the difference between the observer frequency and the centre of the frequency sweep of the broadband shaped pulses. For the optimisation measurements, the frequency of the observer channel was fixed and the frequency of the pump pulse was changed stepwise. We shifted the magnetic field with the pump pulse so that we always pumped on the maximum of the spectrum (see S1). During the increase of the offset, the position of the observer pulses in the spectrum will change as the spectrum is shifted with the pump pulse resulting in a decrease of the echo for higher offsets. Table 2 shows an overview over all tested pump pulse parameter sets.

**Table 2:** The parameters for the broadband shaped pump pulses.

| Pulse type | Length [ns] | Frequency width [MHz] | Offset [MHz] | additional parameter |
|---|---|---|---|---|
| **chirp** | 100 | 80, 120, 160 200 | 70-130 | $t_\mathrm{r} = \frac{t_p}{4}, 10\text{ ns}, 30\text{ ns}$ |
| **short chirp** | 36 | 80, 120,160, 200 | 70-130 | $t_\mathrm{r} = \frac{t_p}{4}, 10\text{ ns}, 30\text{ ns}$ and without quarter sine smoothing |
| **WURST** | 100 | 80, 120, 160, 200 | 70-130 | $n = 6, 12, 24$ |
| **HS{1,1}** | 100 | 50, 70, 90, 110 | 70-130 | $\beta = 4/t_p, 6/t_p, 8/t_p, 10/t_p$ |
| **HS{1,6}** | 100 | 50, 70, 90, 110 | 70-130 | $\beta = 4/t_p, 6/t_p, 8/t_p, 10/t_p$ |

We used the same parameters for the observer pulses as before, meaning that we tried rectangular and Gaussian observer pulses at a microwave frequency of 33.91 GHz and 33.93 GHz at 100 % and 60 % amplitude, respectively, (see Tables S1 and S2) and combined them with all the broadband shaped pulses from Table 2. This results in 504 different settings (Table 2) for the pump pulse and 8 different settings for the observer pulses, which gives a total of 4032 different DEER settings. As the measurement of full DEER traces and subsequent determination of the MNR would be very time consuming we used the $\eta_{2P}$ parameter as an estimation for the MNR. This was suggested by (Doll et al., 2015) and already used by other authors (Spindler Philipp E. et al., 2013; Tait and Stoll, 2016). As it requires only two points of the DEER trace, the measurement time can be drastically reduced. However, it has the disadvantage that artefacts, e.g. echo crossing artefacts or nuclear modulation might remain undetected. Therefore, we decided to additionally perform phase cycling and nuclear modulation averaging. For different observer pulse settings, the $\eta_{2P}$ parameters are not necessarily comparable, because $\eta_{2P}$ assumes a constant absolute noise level. However, this noise level could change with different integration windows. Hence, we identified the best chirp, WURST, HS{1,1} and HS{1,6} pulse for each observer pulse setting and recorded full DEER traces of them, giving a total number of 16 traces of the type RS and GS each.

Exemplary heat maps showing the $\eta_{2p}$ for Gaussian observer pulses and 100 ns pump pulses can be found in Fig. 5. We could identify several trends that were true for all observer pulse settings. HS{1,1} and HS{1,6} pulses have higher maximum $\eta_{2p}$ values than chirp and WURST pulses. HS{1,1} and HS{1,6} have their highest $\eta_{2p}$ values for smaller offsets than chirp and WURST pulses. This fits to the steeper flanks in their excitation profiles and a resulting smaller overlap with the observer frequency. Nonetheless, the overall range of reasonable offsets for all pulses is rather small and within a range of 80 MHz and 100 MHz, meaning that for nitroxide-nitroxide DEER and our setup the width of the spectrum and the resonator profile has a more crucial influence in choosing the right offset than the excitation profiles of the different pump pulses. HS{1,1} and HS{1,6} pulses have smaller ideal frequency widths of 90 MHz and 110 MHz, whereas for chirp and WURST pulses the frequency widths seem to be ideal at 120 MHz and 160 MHz. This fits to the already mentioned observation, that WURST pulses have smaller excitation profiles for a given sweep width with the used parameters than HS{1,1} and HS{1,6} pulses (see Fig. 4). Interestingly, despite their lower adiabaticity the short chirp pulses with a length of 36 ns had a larger $\eta_{2p}$ value than the chirps with a length of 100 ns for all observer pulse settings. Quarter sine smoothing does not necessarily lead to a better performance of the short chirp pulses. For the WURST pulses, a value of $n = 6$ gives the best performance with all observer pulses. For different observer pulses, we find that the best performance of HS{1,1} and HS{1,6} pulses can be achieved with $\beta$ parameters ranging from $6/t_p$ to $10/t_p$.

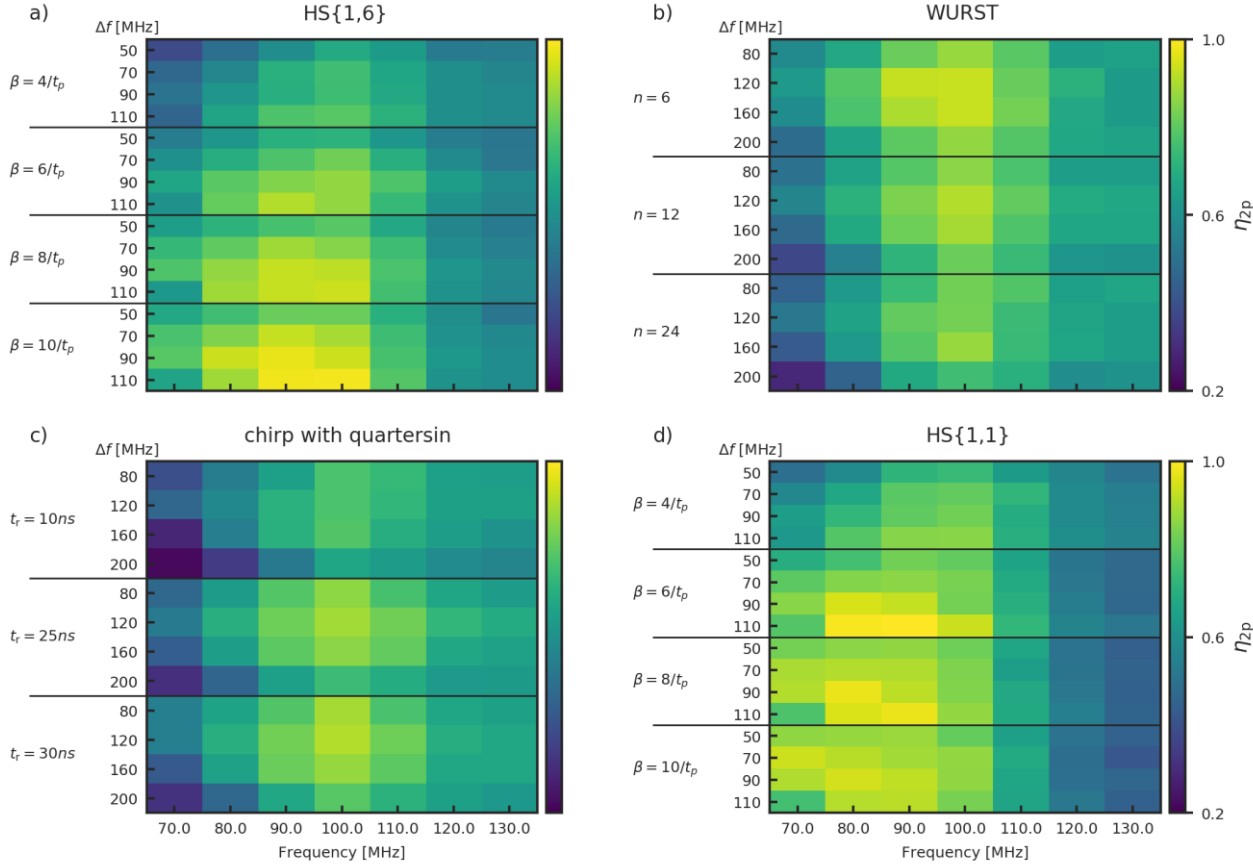

**Figure 5:** Heat maps with the $\eta_{2p}$- values for 4p-DEER measurements with an observer pulse length of 56 ns, Gaussian pulses and an integration window of 56 ns. The observer frequency is at 33.93 GHz. The pump pulse length is 100 ns. Each heat map shows a different pump pulse type: (a) HS{1,6}, (b)WURST , (c) chirp with quarter sine, and (d) HS{1,1}.

For all observer pulse settings, we identified the best parameter set for each pulse family resulting in a maximised $\eta_{2p}$. We then recorded a full DEER trace for each family and compared them by their MNRs. All results for the full DEER traces can be found in Tables S5 and S6. Table 2 shows the parameters and observer pulses that resulted in the best performing chirp, WURST, HS{1,1} and HS{1,6} pulses for the full DEER traces. We found that also for broadband shaped pump pulses

10 Gaussian observer pulses outperform rectangular ones. Again, this hints that Gaussian observer pulses can successfully reduce the frequency overlap with the pump pulse due to their missing sidebands. In all scenarios we found that an observer pulse that is positioned with a 70 MHz offset to the maximum of the resonator profile performs better than an observer pulse position with a 90 MHz offset to the maximum of the resonator profile. The offset to the broadband shaped pump pulse, however, does not change on average, which means that in the former case the observer and pump pulse have a more symmetric positioning

15 around the maximum.

**Table 3:** The parameters of the observer and pump pulse that gave the best MNR for each pump pulse type. All observer pulses are Gaussian pulses with a pulse length of 74 ns for a 60 % intensity and 56 ns for a 100 % intensity. The observer frequency was 33.93 GHz in all cases. The MNR was evaluated up to 7 µs.

| Pump pulse | Obs. Amp. [%] | $t_\pi$ [ns] | $\Delta f$ [MHz] | Offset [MHz] | MNR | Mod depth $\lambda$ |
|---|---|---|---|---|---|---|
| **HS{1,6}** ($\beta = 10/t_p$) | 60 | 100 | 110 | 90 | 45 | 0.61 |
| **WURST** ($n$=6) | 60 | 100 | 160 | 90 | 40 | 0.63 |
| **Chirp** (no smoothing) | 100 | 36 | 120 | 80 | 45 | 0.49 |
| **HS{1,1}** ($\beta = 8/t_p$) | 100 | 100 | 110 | 90 | 50 | 0.52 |

Broadband shaped pump pulses lead to a larger modulation depth as rectangular and Gaussian pulses. Whereas for non-broadband pulses the modulation depth is limited to around 30 % with our setup, we achieved an increase of up to 63 % with WURST pulses. HS{1,6} pulses also lead to high modulation depths of 61 %. For chirp and HS{1,1} pulses smaller modulation depths of approx. 50 % were observed. However, the highest modulation depth will not necessarily lead to the highest MNR as can be seen in Table 3. This is due to a larger background decay of pulses with a higher inversion efficiency and will be

analysed in the next section. Due to a higher bandwidth overlap, broadband shaped pulses will also reduce the echo intensity stronger than rectangular or Gaussian pulses. HS{1,1} pulses seem to be a good compromise between a high modulation depth, a high echo intensity and a background decay that is not too steep. They resulted in the highest MNR of 50 with a pulse length of 100 ns, an offset of 90 MHz, a frequency bandwidth of 110 MHz and $\beta = 8/t_p$ with the observer pulses being Gaussian pulses with an amplitude of 100 % and a frequency of 33.93 GHz. Interestingly, this performance is achieved although the

broadband pulse does not achieve a complete inversion (Fig. S8d). The modulation depth in that case increased to 52 % (Fig. S11). This corresponds to an MNR increase of 43 % compared to RR and 22 % compared to GG. To estimate the lower limit of distances that can be determined with such a 100 ns pulse, we performed a simulation to see when the spins are actually flipped during the experiment (see S7). A visual inspection reveals that most spins are flipped between 20 ns and 80 ns within the pulse duration, making it an effective length of 60 ns where the spins flips occurs, which would correspond to a minimum

detectable distance limit of 2.3 nm instead of 2.8 nm for a 100 ns spin flip period.

Depending on the resonator and the microwave amplifier, different $B_1$ field strengths are available on different spectrometers. However, as the inversion efficiency of broadband shaped pulses is less dependent on the $B_1$ field strength as is the case for rectangular and Gaussian pulses, who always require a proper adjustment of the pulse length, we assume the findings here to be rather generalisable. In order to discuss this more quantitatively we simulated inversion profiles of the best performing

pulses from Table 3 for $B_1$ field strengths.

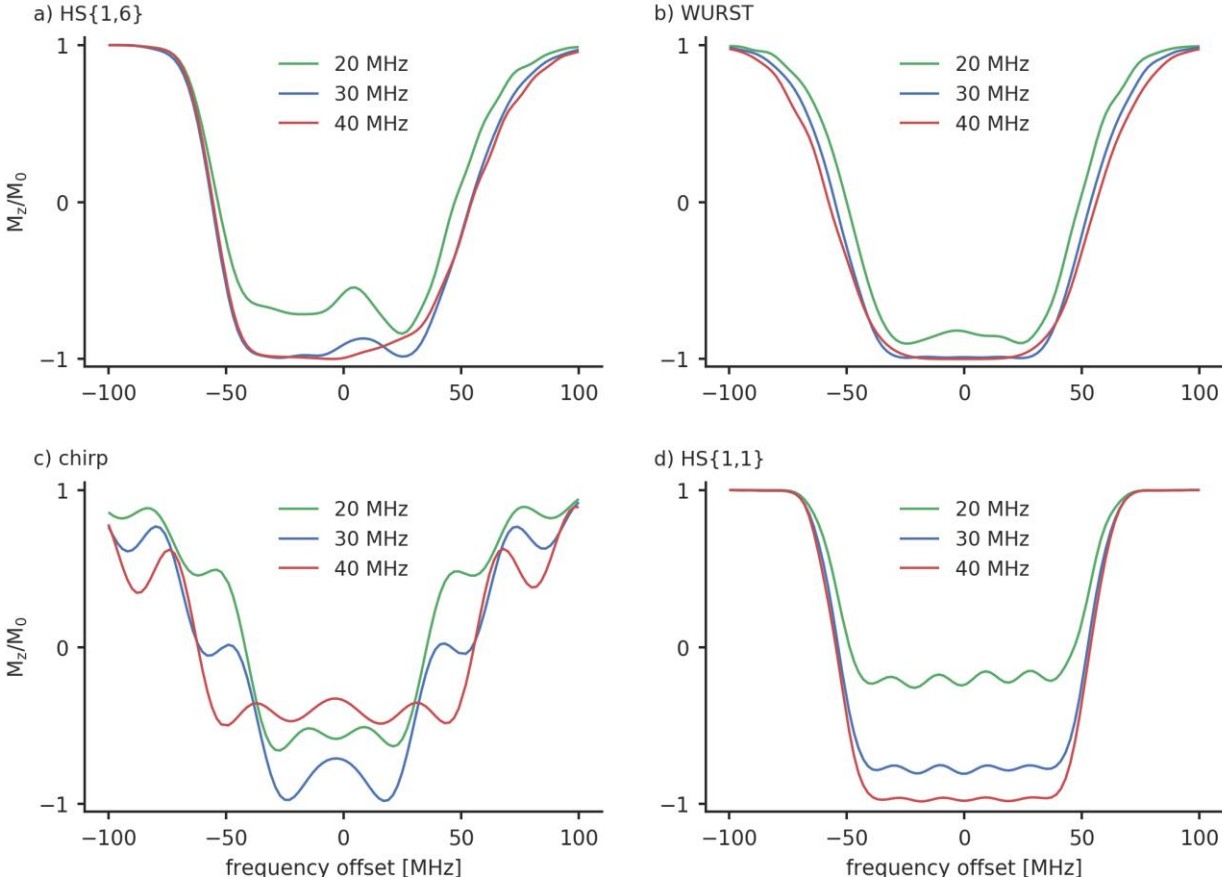

**Figure 6:** The inversion profiles of the best performing (a) HS{1,6}, (b) WURST, (c) chirp and (d) HS{1,1} pulses with the parameters from Table 3. They were simulated with a $B_1$ field strength of 20 MHZ (green), 30 MHz (blue) and 40 MHz (red). These field strengths correspond to a $\pi$-pulse lengths of 25.0 ns, 16.7 ns (which approximately correspond our setup) and 12.5 ns. The $B_1$ field here is depicted as the Rabi frequency.

We compare the pulse profiles with $B_1 = 30$ MHz, which corresponds our setup, with the cases where a lower ($B_1 = 20$ MHz) or higher ($B_1 = 40$ MHz) $B_1$ field strengths are reached. Figure 6 shows how the different pulses behave, when different $B_1$ field strengths are used. The WURST pulse (Fig. 6b) shows the least variation for different $B_1$ field strengths. As expected the inversion efficiency drops a little bit for $B_1 = 20$ MHz. But this drop seems to be rather insignificant and good modulation depths can still be expected. The decrease in inversion efficiency is a bit more significant for the HS{1,6} pulse so that a small reduction in the modulation depth is possible here. Both pulse profiles do not show significant changes when a higher $B_1$ field strength is used. The HS{1,1} pulse has a massive drop in inversion efficiency when going to lower $B_1$ field strengths. This comes not as a surprise as the inversion efficiency is already incomplete at $B_1 = 30$ MHz. Here, it might be advantageous to reduce the $\beta$ parameter of the HS{1,1} pulse. As it has been stated earlier this will increase the inversion efficiency. For a higher $B_1$ field strength of $B_1 = 40$ MHz the inversion efficiency of this HS{1,1} will increase. Therefore, a higher modulation depth comparable to the HS{1,1} pulse is expected. As this will also increase the background decay, a higher MNR is not guaranteed. The chirp pulse also shows a rather strong decrease in the inversion efficiency for a $B_1 = 20$ MHz. However, the inversion efficiency also decreased for a higher $B_1$ field strength of $B_1 = 40$ MHz. This rather unexpected behaviour is probably caused by an insufficient smoothing of the edges of the chirp pulse. With higher $B_1$ field strength the initial effective magnetic field vector in the accelerated frame becomes less aligned with the z-axis. Therefore, smoothing becomes more important. In Fig. S10, we compared the inversion profiles of 36 ns and 100 ns chirp pulses with and without quarter sine smoothing. When quarter sine smoothing is applied, chirp pulses can with a length of 36 ns indeed reach a high inversion efficiency with $B_1 = 40$ MHz. (Fig. S10b). As the width of the inversion profile of this chirp pulse drops significantly for smaller $B_1$ field

strengths, it is only advisable to implement a quarter sine smoothing with chirp pulses of a length of 36 ns when enough microwave power is available. The situation looks different for chirp pulses with a pulse length of 100 ns. Here, the inversion profile looks very similar for all tested $B_1$ field strengths. Particularly for smaller $B_1$ field strengths we expect 100 ns chirp pulses to outperform chirp pulses with a length of 36 ns.

Another crucial parameter for DEER measurements that can vary from setup to setup is the width of the resonator profile. Here, we have a FWHM of approximately 200 MHz. Larger widths do not seem to be necessary because they would exceed the width of the spectrum of the nitroxide. If only a smaller width is available, the offset between pump and observer pulses might need to be reduced. This would increase the overlap between the observer and pump pulses. This problem could be overcome by either using longer pump pulses or reducing the frequency width of the broadband shaped pulses. As a narrower

resonator profile is also necessarily steeper, it might also be necessary to perform a resonator bandwidth compensation as suggested by (Doll et al., 2013). Performing a resonator bandwidth compensation with our setup does not give a significant advantage in the $\eta_{2p}$ value (see S15). This is probably due to the rather flat resonator profile in the region with maximum sensitivity where the pump pulse is applied.

### 3.3 Background behaviour

The broader excitation profile of broadband shaped pulses will increase the background decay which results in a higher noise level of broadband shaped pump pulses compared to rectangular or Gaussian pump pulses. We find an approximately linear relation between the modulation depth and the background decay (see S19). To investigate this effect more deeply we evaluated the MNR of the experimental DEER form factors excluding the later part of the form factor and only taking into account the first part up to a truncation time $\tau_{\text{truncation}}$. Truncation the DEER trace will not change the modulation depth, but due to the

background decay, the noise level will be different. Figure 7 shows the MNR of broadband pulses as a function of the truncation time $\tau_{\text{truncation}}$. As expected, the MNR decreases with increasing $\tau_{\text{truncation}}$ for all pulses, because of the increase of the noise. However, the rate of the decrease in MNR is different for different pulse types, which means that the relative performance of the pulses also depends on the length of the DEER trace and therefore on the distance between the spin centres that is supposed to be measured.


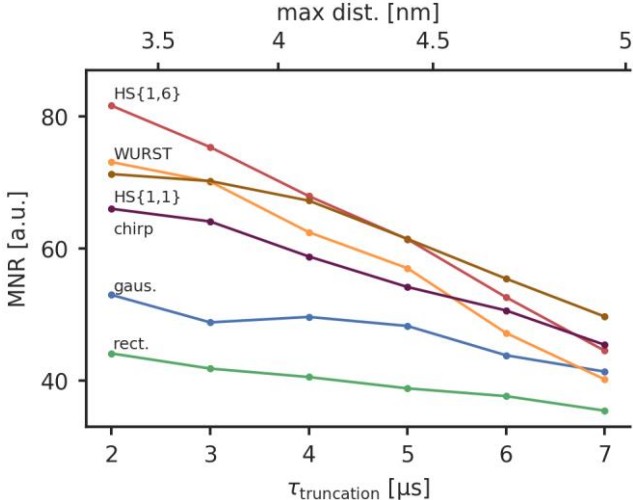

**Figure 7:** The MNR value as a function of the dipolar evolution time up to which the noise has been evaluated, i.e. the truncation time $\tau_{\text{truncation}}$. The sample has a concentration of 80 µM of spin-labelled ligand. The maximum distance according to equation (13) is depicted in the upper x-axis. The line between the points is only a guidance for the eyes.

It turns out that HS{1,6} and WURST pulses with their higher modulation depths have the highest MNR for short DEER traces, whereas for longer traces HS{1,1} and chirp pulses are better. The background decay seems to play a decisive role for the MNR and the pulses resulting in a high modulation depth also have a larger background decay. As the background decay causes the noise level to increase with increasing dipolar evolution time, its influence is less pronounced for short DEER traces, where the high modulation depth seems to be leading to a high MNR. For longer traces, a high modulation depth is linked to a strong background decay and a high noise level towards the end of the trace. Therefore, the MNR of pump pulses generating a high modulation depth decreases stronger than for pulses effecting a smaller modulation depth. This means that HS{1,1} and chirp pulses perform better for longer traces.

As rectangular and Gaussian pump pulses have rather small modulation depths, the corresponding decrease of the MNR due to the background decay is also rather small, that means that the improvement achievable with broadband shaped pulses is greater for shorter DEER traces. For short truncation times $\tau_{\text{truncation}}$ of 2 μs we observe an increase in MNR from 44 for rectangular pulses (RR) to 82 for the best broadband shaped pulse (RS), which was an HS{1,6} pulse in this case. This corresponds to an increase of 86 %. For long truncation times $\tau_{\text{truncation}}$ of 7 μs, this increase goes down to 43 %. This means that the MNR improvement that can be achieved by broadband shaped pulses can be drastically dependent on the length of the measured DEER trace and therefore on the distance range to be covered by the measurement. For a concentration of 80 μM, a high MNR improvement can be achieved if the maximum distance of interest is below 4 nm with pulse that achieves a high modulation depth. This would correspond to the HS{1,6} and WURST pulse in this case. If longer distances up to 5 nm shall be detected, it seems to be advantageous to use pulses that might not give the highest modulation depth in order to reduce the background decay. An extrapolation for higher truncation times shows that if even longer distances are of interest, broadband shaped pulses will not give a better MNR compared to rectangular pulses. Here, it is necessary to reduce the background decay by using lower concentrations.

The performance of all the pulses at $\tau_{\text{truncation}} = 2$ μs can be found in Tables S7-S10. The chirp, WURST, HS{1,1} and HS{1,6} pulse resulting in the best MNR are summarized in Table 4. For the broadband shaped pulses there were some minor changes in the parameters that gave the MNR when the truncation time was set to a shorter value of $t_{\text{truncation}} = 2$ μs. For RR and GG there were changes in the best parameter settings.

**Table 4:** The parameters of the observer and pump pulse resulting in the best MNR for each pulse type when the MNR was evaluated up to $\tau_{\text{truncation}} = 2$ μs. All observer pulses are Gaussian pulses with a pulse length of 74 ns for 60 % intensity and 56 ns for 100 % intensity. The observer frequency was 33.93 GHz for all pulses.

| Pump pulse | Obs. Amp. [%] | $t_\pi$ [ns] | $\Delta f$ [MHz] | Offset [MHz] | MNR | Mod depth $\lambda$ |
|---|---|---|---|---|---|---|
| **HS{1,6}** ($\beta = 10/t_p$) | 60 | 100 | 110 | 90 | 82 | 0.61 |
| **WURST** ($n$=6) | 100 | 100 | 160 | 90 | 73 | 0.63 |
| **Chirp** (no smoothing) | 100 | 36 | 120 | 80 | 65 | 0.49 |
| **HS{1,1}** ($\beta = 8/t_p$) | 60 | 100 | 110 | 80 | 74 | 0.52 |

### 3.4 Concentration dependence

To check for a concentration dependent performance of broadband shaped pulses we also prepared a sample with a lower concentration of 30 μM ligand and 60 μM WGA and performed DEER measurements with the optimised parameter settings

for the short chirp, WURST, HS{1,1} and HS{1,6} pulses. We did, however, not check observer frequencies of 33.91 GHz, since they always performed worse than an observer position of 33.93 GHz. For RR we tested an offset of 70 MHz and 60 % intensity, for GG we tested an offset of 70 MHz as well, but an intensity of 100 %, as these settings performed best before.

This sample showed almost no background for all used pump pulses (see SI 12). As the influence of the background is minimised due to the low concentration we expected to find the trends in the MNR as for the case of the high concentrated samples and short truncation times. Figure 8 shows the MNR as function of the truncation time point $\tau_{\text{truncation}}$ up to which the noise has been evaluated. As expected, no significant decrease of the MNR with higher truncation times $\tau_{\text{truncation}}$ was found. Without a significant background the noise towards the end of the background-corrected form factor does not increase significantly. The decrease of the MNR found for the high concentration sample was therefore not observed here. For some pulses there is a slight increase in the MNR with the truncation time, however, we assigned this behaviour to a numerical uncertainty in the analysis.

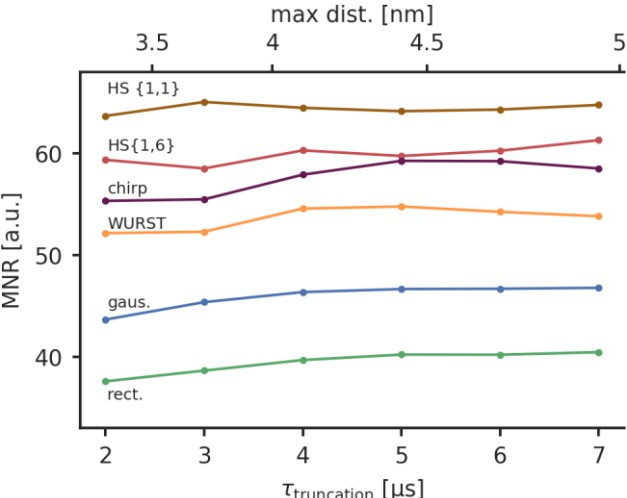

**Figure 8:** The MNR as a function of the dipolar evolution time up to which the noise has been evaluated. The sample has a concentration of 30 µM of spin-labelled ligand. The maximum distance according to equation (13) is depicted in the upper x-axis. The line between the points is only a guidance for the eyes.

**Table 5:** The parameters of the observer and pump pulse resulting in the best MNR for each pulse type. All observer pulses are Gaussian pulses with pulse lengths of 74 ns for 60 % intensity and 56 ns for 100 % intensity. The observer frequency was 33.93 GHz for all pulses. The MNR was evaluated up to 7 µs.

| Pump pulse | Obs. Amp. [%] | $t_\pi$ [ns] | $\Delta f$ [MHz] | Offset [MHz] | MNR | Mod depth $\lambda$ |
|---|---|---|---|---|---|---|
| **HS{1,6} ($\beta = 10/t_p$)** | 100 | 100 | 110 | 90 | 61 | 0.55 |
| **WURST ($n$=6)** | 60 | 100 | 160 | 100 | 54 | 0.59 |
| **Chirp** (no smoothing) | 100 | 36 | 120 | 80 | 58 | 0.46 |
| **HS{1,1} ($\beta = 8/t_p$)** | 100 | 100 | 110 | 90 | 65 | 0.47 |

With the low concentration sample, we found an MNR of 40 and a modulation depth of 31 % for rectangular pulses, for Gaussian pulses we found an MNR of 47 and a modulation depth of 30 %. Thus, also at lower spin concentrations the Gaussian

pulses lead to a similar modulation depth as rectangular pulses, but again to an overall higher MNR. Table 5 shows the results for the different broadband shaped pump pulses in combination with the observer pulse with whom they performed best.

Table 5 shows the optimised parameters for the different pump pulses. All results can be found in Table S11. The parameters found for the observer and pump pulses differ slightly from the parameters identified for the high concentration sample, but lie in a similar range.

The broadband shaped pump pulses resulted in a modulation depth that is a bit lower than for the sample with the high concentration. The MNR was lower as well. Furthermore, the order of performance of the different pulse types changed. While we expected HS{1,6} and WURST pulses with their high modulation depths to perform better than HS{1,1} and chirp pulses for a sample less susceptible to background influence, HS{1,1} pulses were actually performing best and WURST pulses were the worst broadband shaped pulses. HS{1,1} pulses lead to an increase in the MNR of 60 % compared to rectangular pulses. This is also lower than the 86 % increase that was obtained for the 80 µM ligand concentration. The reason for the change of this behaviour is probably a difference in the resonator profile that we noticed compared to the other sample with the higher concentration (see S22). The achieved $B_1$ field was a bit lower for this sample which changes the performance of the pulses. However, HS{1,1} and HS{1,6} pulses both give a good MNR with a high concentration as well as with a low concentration.

When the MNR shall be increased by using broadband shaped pulses to detect long distances > 5 nm, lower concentrations are preferable as they reduce the enhancement of the background decay. Here, switching to a concentration of 30 µM of the doubly labelled ligand was enough the significantly reduce the influence of the background. In S19 we performed analytical calculations to estimate the potential MNR increase that can be achieved by switching to broadband shaped pulses for different concentrations and distance ranges. For maximum distances below 4 nm an increase of the MNR can be expected for all concentrations up to approximately 100 µM. The situation is different if distances over 6 nm shall be detected. A significant gain can only be expected for smaller concentrations in the range between 10-30 µM. For higher concentrations the MNR gain drops quickly. For higher concentrations in the range of 80 µM a MNR decrease has to be expected in this distance regime. This is discussed in more detail in S21.

**4 Conclusion and Outlook**

We have compared various broadband shaped pulses as pump pulses for DEER spectroscopy in Q-band performed on samples with nitroxide spin labels and investigated under which circumstances they perform best. By increasing the inversion profile, broadband shaped pulses can increase the modulation depth from 30 % with rectangular pulses up to 60 %. However, with a larger inversion profile of broadband shaped pulses the overlap with the observer pulse and the background decay will also increase. Both of those effects will tend to reduce the MNR. The overall MNR increase will therefore be a compromise between the increase in the modulation depth and the smaller echo and larger background contribution.

Systematic analysis of a trial-and-error optimisation has yielded that the performance of broadband shaped pulses depends on the dipolar evolution time and the concentration of spin centres. Larger dipolar evolution times mean that the background has decayed stronger by the end of the form factor. Pulses with a higher inversion efficiency will produce a larger background decay and their performance decreases stronger for longer traces than for pulses with a smaller inversion efficiency. We found HS{1,1} and HS{1,6} in combination with Gaussian observer pulses to give a good MNR for high as well as low spin concentrations. HS{1,1} have a lower inversion efficiency and therefore a lower modulation depth but they perform better with longer traces needed for longer distances. The exact parameters depend on the setup, but with values of $\beta = 8/t_p$ or $\beta = 10/t_p$, $t_p = 100$ ns, $\Delta f = 110$ MHz and an offset of 80 MHz or 90 MHz we typically achieved good results. If a high modulation depth which is particularly suitable for short distances should be achieved, HS{1,6} and WURST pulses are the

best pulses. Good parameters are $\beta = 10/t_p$, $t_p = 100$ ns, $\Delta f = 110$ MHz and an offset of 90 MHz for HS{1,6} and $n = 6$, $t_p = 100$ ns, $\Delta f = 160$ MHz and an offset of 90 MHz or 100 MHz for WURST pulses.

## 5 Data availability

The raw data can be downloaded at https://doi.org/10.5281/zenodo.3726735 (Scherer et al., 2020).

## 6 Author contribution

AS, ST, SW and MD conceived the research idea and designed the conducted experiments. AS conducted the EPR experiments and analysed the results with ST. The spin labelled ligand was synthesised in the lab of VW. AS prepared all the figures and wrote the draft manuscript. All authors discussed the results and revised the manuscript.

## 7 Competing interests

The authors declare no conflict of interests.

## 8 Acknowledgements

We thank Philipp Rohse for the synthesis of the spin labelled ligand. Jörg Fischer is thanked for sample preparation. This project has received funding from the European Research Council (ERC) under the European Union's Horizon 2020 research and innovation programme (Grant Agreement number: 772027 — SPICE — ERC-2017-COG). A.S. gratefully acknowledges financial support from the Zukunftskolleg of the University of Konstanz. A.S., S.T. and S.W. gratefully acknowledge financial support from the Konstanz Research School Chemical Biology (KoRS-CB).

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
