# Peer review of "Optimising broadband pulses for DEER depends on concentration and distance range of interest"

_Magnetic Resonance, 2020_

## Referee Comment (RC1) · Thomas Prisner (Referee) · 9 Mar 2020

This manuscript tries to tackle experimentally the question how the pump efficiency of broadband microwave pulses influences the quality of DEER data. Broadband mw pulses become available for pulsed EPR experiments by fast arbitrary waveform generators recently, allowing to adopt broadband excitation pulses known since a long time in the field of NMR for EPR experiments. For DEER experiments it has been shown that broadband pulses achieve larger modulation depth of the dipolar coupled bi-radical sample. On the other hand the larger pump efficiency also leads to a stronger decay arising from intermolecular interactions to the other spin labeled molecules in the

sample. In this manuscript the interplay of modulation depth and decay arising from a series of broadband pulses (sech/tanh, WURST, linear chirp with rectangular amplitude shape) have been compared to excitation by rectangular or Gaussian monochromatic pulses. The experimental work performed at a commercial Q-band spectrometer is of good quality and compares the effect of different pump pulses with respect to a MNR (modulation to noise ratio) merit function. This function is defined as the ratio of the modulation depth to the over the time trace averaged noise level. The later part of this function depends on the length of the observer time window, because the noise which is constant per time in the original time trace, increases by the division procedure usually used to remove the intermolecular part. This problem has already been discussed in a recent paper by Fábregas et al., where other procedures to obtain distance distributions from the original data were proposed. Nevertheless, because these other simulations methods are not standard so far, the approach discussed here to optimize the experimental time traces is important and worthwhile to be published after several issues are addressed. 1) For me it is not perfectly clear if the merit function defined here (MNR) is really the most important point. Usually the distance information is encoded in the first part of the time trace and the longer times are only necessary to fit well the intermolecular background function – a necessary procedure to obtain reliable distance information. The question is how large the increasing noise at the end of the time trace is important for this purpose. This point should be discussed and clarified. 2) The manuscript talks about the intermolecular background function but for the larger spin concentration, where most of the experiments are performed and most of the conclusions are taken from, the original time traces including this background function are not shown! This has to be included! It is not enough to show the background density k as in Figure S11. 3) In Figure S10 an unexpected large suppression of the echo intensity by longer broadband pump pulses is shown. This is totally unexpected for the given bandwidth of these pulses and contrary to own experiences, where longer pulses show better frequency shapes! If the pulses are generated by the Bruker software some care has to be taken to use the right amplitude setting, especially when

the frequency runs over the carrier frequency (from minus offsets to plus offsets). The pulse profiles shown in Figure S7 are probably only calculated pulses and not really measured ones? Experimentally recording them including the resonator profile (which also seems somewhat suspicious to me) might give some hints on what is going on here! This issue is rather important for the conclusions drawn here from the shaped pulses! 4) The concentration dependence of the behavior is only discussed in a rather trivial and non-quantitative manner, despite the fact that it showed to be the major parameter influencing the improvement by the broadband pulses (comparison Figure 6 and 7). That lower concentrations of spins are advantageous, especially for larger distances or broader distance distributions is well known in the community. Because broadband pulses might be especially interesting for these kind of systems, this should be discussed more quantitatively! The discussion in the SI including Figure S13 and the text after it is only very qualitative and rather trivial. 5) Minor point: In the supporting information equation (2) is wrong. After that there is a spelling error (ration).

---

## Referee Comment (RC2) · Anonymous Referee #2 · 17 Mar 2020

This work investigates in detail the dependence of DEER signal-to-noise ratio on many experimental parameters (pulse shapes, pulse lengths, pump-observer separation, concentration, truncation).

This work does not introduce any new method. Hyperbolic-secant pump pulses and Gaussian observer pulses are already commonly used in 4-pulse DEER, and many of the optimizations and explorations presented are routinely done by spectroscopists.

Although not innovative for this reason, this manuscript demonstrates explicitly that it can be beneficial to spend time to optimize experimental parameters carefully. Novices to the field will find this manuscript particularly instructing as a tutorial.

I recommend publication in Magnetic Resonance, after the comments below are addressed.

[1] In section 3.3 (background decay), examine in detail how the type of shaped pulse affects the MNR if the recently published superior background correction method (kernel inclusion) is used. It is crucial to include this in this work.

[2] Discuss in more detail whether and how the findings in this work are applicable to other samples (different distance distributions, different concentrations) and spectrometers (different resonator profiles, different Tx fidelity). From the current manuscript, it is unclear whether the findings are generalizable. This is important, since it appears to be the purpose of the manuscript to make some general statements about experimental settings in DEER.

[3] - Eq.(3): Specify that the time axis is defined such that t=0 at the center of the pulse. - Eq.(11):  $(k^*|t|)^{(d/3)}$  instead of  $k^*t^{(d/3)}$  - Eq.(12): A factor of 2 might be missing. - 8.13: "i.e. a chirp pulse" - 12.18: Here, it is not clear how the numbers for the minimum detectable distance limit are obtained. - Kupce needs a grave accent on the c. Bohlen needs an umlaut on the o. - SI Eq.(2): t\_truncation instead of tau\_2

---

## Referee Comment (RC3) · Anonymous Referee #2 · 3 Apr 2020

As stated in my initial comments, I don't see anything scientifically novel in this work. It's a tutorial possibly useful for novices. As MR is a new journal, I am uncertain as to how such contributions align with the journal's aims and scope.

The author addressed all my comments.

---

## Author Response (AR1)

**1. Referee comments**

**1.1. Thomas Prisner**

This manuscript tries to tackle experimentally the question how the pump efficiency of broadband microwave pulses influences the quality of DEER data. Broadband mw pulses become available for pulsed EPR experiments by fast arbitrary waveform generators recently, allowing to adopt broadband excitation pulses known since a long time in the field of NMR for EPR experiments. For DEER experiments it has been shown that broadband pulses achieve larger modulation depth of the dipolar coupled bi-radical sample. On the other hand the larger pump efficiency also leads to a stronger de-cay arising from intermolecular interactions to the other spin labeled molecules in the sample. In this manuscript the interplay of modulation depth and decay arising from a series of broadband pulses (sech/tanh, WURST, linear chirp with rectangular amplitude shape) have been compared to excitation by rectangular or Gaussian monochromatic pulses. The experimental work performed at a commercial Q-band spectrometer is of good quality and compares the effect of different pump pulses with respect to a MNR(modulation to noise ratio) merit function. This function is defined as the ratio of the modulation depth to the over the time trace averaged noise level. The later part of this function depends on the length of the observer time window, because the noise which is constant per time in the original time trace, increases by the division procedure usually used to remove the intermolecular part. This problem has already been discussed in a recent paper by Fábregas et al., where other procedures to obtain distance distributions from the original data were proposed. Nevertheless, because these other simulations methods are not standard so far, the approach discussed here to optimize the experimental time traces is important and worthwhile to be published after several issues are addressed. 1) For me it is not perfectly clear if the merit function defined here (MNR) is really the most important point. Usually the distance information is encoded in the first part of the time trace and the longer times are only necessary to fit well the intermolecular background function – a necessary procedure to obtain reliable distance information. The question is how large the increasing noise at the end of the time trace is important for this purpose. This point should be discussed and clarified.2) The manuscript talks about the intermolecular background function but for the larger spin concentration, where most of the experiments are performed and most of the conclusions are taken from, the original time traces including this background function are not shown! This has to be included! It is not enough to show the background density k as in Figure S11. 3) In Figure S10 an unexpected large suppression of the echo intensity by longer broadband pump pulses is shown. This is totally unexpected for the given bandwidth of these pulses and contrary to own experiences, where longer pulses show better frequency shapes! If the pulses are generated by the Bruker soft-ware some care has to be taken to use the right amplitude setting, especially when the frequency runs over the carrier frequency (from minus offsets to plus offsets). The pulse profiles shown in Figure S7 are probably only calculated pulses and not really measured ones? Experimentally recording them including the resonator profile (which also seems somewhat suspicious to me) might give some hints on what is going on here! This issue is rather important for the conclusions drawn here from the shaped pulses! 4) The concentration dependence of the behavior is only discussed in a rather trivial and non-quantitative manner, despite the fact that it showed to be the major parameter influencing the improvement by the broadband pulses (comparison Figure6 and 7). That lower concentrations of spins are advantageous, especially for larger distances or broader distance distributions is well known in the community. Because broadband pulses might be especially interesting for these kind of systems, this should be discussed more quantitatively! The discussion in the SI including Figure S13 and the text after it is only very qualitative and rather trivial. 5) Minor point: In the supporting information equation (2) is wrong. After that there is a spelling error (ration).

**1.2 Referee 2**

This work investigates in detail the dependence of DEER signal-to-noise ratio on many experimental parameters (pulse shapes, pulse lengths, pump-observer separation, concentration, truncation).This work does not introduce any new method. Hyperbolic-secant pump pulses and Gaussian observer pulses are already commonly used in 4-pulse DEER, and many of the optimizations and explorations presented are routinely done by spectroscopists. Although not innovative for this reason, this manuscript demonstrates explicitly that it can be beneficial to spend time to optimize experimental parameters carefully. Novices to the field will find this manuscript particularly instructing as a tutorial. I recommend publication in Magnetic

Resonance, after the comments below are ad-dressed.[1] In section 3.3 (background decay), examine in detail how the type of shaped pulse affects the MNR if the recently published superior background correction method (kernel inclusion) is used. It is crucial to include this in this work.[2] Discuss in more detail whether and how the findings in this work are applicable to other samples (different distance distributions, different concentrations) and spectrometers (different resonator profiles, different Tx fidelity). From the current manuscript, itis unclear whether the findings are generalizable. This is important, since it appears to be the purpose of the manuscript to make some general statements about experimental settings in DEER.[3] - Eq.(3): Specify that the time axis is defined such that t=0 at the center of the pulse.- Eq.(11): $(k*|t|)^{(d/3)}$ instead of $k*t^{(d/3)}$ - Eq.(12): A factor of 2 might be missing. -8.13: "i.e. a chirp pulse" - 12.18: Here, it is not clear how the numbers for the minimum detectable distance limit are obtained. - Kupce needs a grave accent on the c. Bohlen needs an umlaut on the o. - SI Eq.(2): t_truncation instead of tau_2

**2. Response to Referee comments**

**2.1 Response to Thomas Prisner**

5      Dear Thomas,

Thank you very much for your effort reading our manuscript so carefully and for your valuable advices. We strongly believe that the suggested changes and additions improved the manuscript a lot. Please find below a point-by-point reply to all your recommendations (in blue).

10    1) For me it is not perfectly clear if the merit function defined here (MNR) is really the most important point. Usually the distance information is en-coded in the first part of the time trace and the longer times are only necessary to fit well the intermolecular background function – a necessary procedure to obtain reliable distance information. The question is how large the increasing noise at the end of

15    the time trace is important for this purpose. This point should be discussed and clarified.

We have added an additional chapter in the SI where we discuss the MNR as a function of merit and up to which point of the DEER trace it has to been taken into account.

**S2 The MNR as the function of merit**

20    Here, we want to discuss whether the MNR is a suitable function of merit for the determination of distance distributions and up to which time point in the DEER trace, the MNR needs to be evaluated to serve this purpose. Therefore, we performed simulations with a model distance distribution $p_0$ that is based on the narrow distance distribution of the model system used in this study. We approximated the experimentally obtained distance distribution with a Gaussian with a mean at 5.08 nm and a standard deviation of 0.08 nm. We varied the background density in ten steps from $k = 0.01$ 1/µs to $k = 0.3$ 1/µs in combination

25    with a low, medium and high noise level (noise $\sigma_0 = 0.02, 0.05$ and $0.1$) that was added to the DEER trace. The background dimension was set to $d = 3$ and a modulation depth of 0.5 was used. The DEER traces were simulated in the time domain up to 8 µs. For each parameter set we generated ten different traces. To compare the background correction by division (Jeschke et al., 2006) with the kernel inclusion approach as described in (Fábregas Ibáñez and Jeschke, 2020) we analysed all simulated DEER traces with both methods. We did not fit the background but used the true background function. The regularisation

30    parameter was chosen according to the generalised cross-validation method. The quality of the resulting distance distributions $p$ was estimated by the Euclidian distance $D$ from the true distance $p_0$:

$$D(p, p_0) = \|p - p_0\|_2 \tag{1}$$

The MNR of the form factor $F$ was calculated as described in the main text up to a limit of 7 µs according to equation (13) of the main text.

[Figure]

**Figure S2:** The Euclidian distance $D$ of the real and calculated distance distribution as defined in equation (1) is plotted as a function of the MNR. Each dot represents a simulated DEER trace with either low ($\sigma_0 = 0.02$, green), medium ($\sigma_0 = 0.05$, blue) and high ($\sigma_0 = 0.1$, red) noise. The background correction was performed by (a) dividing the DEER trace by the background and (b) including the background in the kernel.

In Fig. S2, the quality of the determined distance distribution was plotted as a function of the determined MNR for both a background correction by division (Fig. S2a) and a kernel inclusion approach (Fig. S2b). For each noise level the MNR only depends on the density of the background as all other parameters are kept constant and only the background density is varied. So a lower MNR corresponds to a higher background density rate and vice versa. For the low noise level ($\sigma_0 = 0.02$), the quality of the determined distance distributions only varies a little for different background density rates. For medium ($\sigma_0 = 0.05$) and high ($\sigma_0 = 0.1$) noise levels, however, the dependency of the quality of the determined distance distribution decreases significantly with a decreasing MNR. If the MNR is only evaluated up to an early point of the form factor, the information of the background decay rate is lost in this case and is not properly included in the MNR as the MNR would then depend nearly exclusively on the given noise level.

[Figure]

**Figure S3:** An exemplary distance distribution obtained for a medium noise level ($\sigma_0 = 0.02$) with (a) a low background density ($k = 0.01$ MHz) and (b) a high background density ($k = 0.3$ MHz). The grey area shows the area that is covered by the calculated distance distribution for ten exemplary DEER traces. The mean of the shaded area is drawn in blue and the true distance is drawn in green.

A closer inspection reveals that whereas the obtained distance distributions for high background densities reproduce the mean of the distance distribution correctly, they overestimate the width of the distribution and the distance appears to be broader as it is (see Fig. S3 for an exemplary data set). Depending on the information that shall be obtained by the DEER measurements, the mean of the distance distribution might suffice. However, if high resolution distance distributions shall be obtained, it seems to be important to optimise the MNR up to the limit which is given by equation (13) of the main text. The comparison of both background correction methods shows that the kernel inclusion gives better results particularly for a high noise and a high background decay. It should therefore be considered as the superior method. However, the correlation between the quality of the determined distance distribution and the MNR is still valid. This is why, we consider the MNR as a proper function of merit, even if the kernel inclusion approach is used.

For a more comprehensive study, the effect of the MNR on the quality of the obtained distance distribution could also be tested for distance distributions with different distance ranges and widths. Such a detailed study was, however, beyond the scope of the this manuscript.

2) The manuscript talks about the intermolecular background function but for the larger spin concentration, where most of the experiments are performed and most of the conclusions are taken from, the original time traces including this background function are not shown! This has to be included! It is not enough to show the background density k as in Figure S11.

We have added a new figure S17 with full DEER traces and background functions of both samples for rectangular and Gaussian pulses.

**S20 Background decay of the DEER traces**

[Figure]

**Figure S17:** The (normalised) experimental raw data of the sample with a 80 µM (a) and 30 µM (b) ligand concentration. The settings for GG (green) were performed with a 100 % pulse amplitude and a 70 MHz offset. For GS (blue), the observer pulses were at a frequency of 70 MHz offset from the centre of the resonator. The pump pulses were HS{1,1} pulses, with the parameters $\beta = 8/t_p$, $t_p = 100$ ns, $\Delta f = 110$ MHz and an offset from the observer pulse of 90 MHz. Note that the acquisition time for the sample with lower concentration was longer in order to reach a similar noise level for both cases. The corresponding form factors are depicted in Fig. S11.

3) In Figure S10 an unexpected large suppression of the echo intensity by longer broadband pump pulses is shown. This is totally unexpected for the given bandwidth of these pulses and contrary to own experiences, where longer pulses show better frequency shapes! If the pulses are generated by the Bruker soft-ware some care has to be taken to use the right amplitude setting, especially the frequency runs over the carrier frequency (from minus offsets to plus offsets). The pulse profiles shown in Figure S7 are probably only calculated pulses and not really measured ones? Experimentally recording them including the resonator profile (which also seems somewhat suspicious to me) might give some hints on what is going on here! This issue is rather important for the conclusions drawn here from the shaped pulses!

We thank Thomas Prisner for this hint. We have redone the experiments and added measured inversion profiles of the pulses. We also want to note that the frequency of the calculated pump pulses does not run over the carrier frequency of the spectrometer because the pulse offset to the observer frequency (carrier frequency) is included. The resonator profile looks unusual (increase at the lower frequency end) because the used resonator is a dual mode resonator. In the figure below the resonator profile recorded over a larger range can be seen. Note that this was recorded with Gd and a different microwave power, hence the different values for the nutation frequency.

[Figure]

However, it was not feasible to use the second mode for the DEER measurements because of the limited width of the nitroxide EDFS spectrum. In order to discuss the issue with the 200 ns and 400 ns pump pulses, we have replaced S12 and S13 with the following new chapter, where we have also included measured inversion profiles of the broadband shaped pump pulses.

**S13 The influence of the length of broadband shaped pump pulses**

Tests with broadband shaped pump pulses with pulse lengths of 200 ns and 400 ns showed that they do not lead to an overall performance increase. This is shown here exemplary by comparing the performance of HS{1,1} pump pulses and Gaussian observer pulses (Fig. S12). There are indeed some pump pulses (for example a HS{1,1} pulse with $\beta = 10/t_p$ and $\Delta f = 110$ MHz) that show an improvement with a longer pulse length, however there is no overall gain by using a pump pulse length of 200 ns.

[Figure]

**Figure S12:** HS{1,1} pump pulses of (a) 100 ns and (b) 200 ns length. The observer pulses were Gaussian pulses with 100 % intensity at an observer position with a 90 MHz offset from the centre of the resonator profile and a pulse length of 56 ns for the $\pi$ pulse. The colour bars are normalised to the same value so that both heat maps are comparable.

We noticed that a major problem with longer broadband shaped pump pulses is that the intensity of the echo can be reduced (Fig. S13a). For a pump pulse offset of 90 MHz, the echo intensity at the zero time of the DEER trace is reduced significantly when increasing the pump pulse lengths from 100 ns over 200 ns to 400 ns.

[Figure]

**Figure S13:** The echo at the zero time of the DEER trace. The observer pulses were Gaussian pulses with 100 % intensity at an observer position with a 70 MHz offset from the centre of the resonator profile and a pulse length of 56 ns for the $\pi$ pulse. The pump pulses were HS{1,1} pulses with $\beta = 8/t_p$ and $\Delta f = 110$ MHz. The offset between the pulses is (a) 90 MHz and (b) 130 MHz.

A comparison of the calculated inversion profiles of the respective pulses (Fig. S14a-c) shows that, whereas the 100 ns pulse should lead to an incomplete inversion, a nearly complete inversion can be expected for the longer pulses. Furthermore, the longer pulses should have slightly steeper excitation flanks. Those trends can indeed be found for the measured inversion profiles. There are some deviations of the measured and calculated inversion profiles. The measured inversion profile of the 100 ns pulse shows an increased frequency width compared to the calculated profile. Furthermore, there is bump in the centre of the frequency sweep. The measured inversion profiles of the longer pulses show the expected steep frequency flanks that can also be seen in the simulation. The inversion profiles of the 200 ns and the 400 ns pulses show a small asymmetry around the centre of the frequency sweep. We assign these deviations to instrumental pulse distortions caused by the spectrometer.

[Figure]

**Figure S14:** Calculated and measured inversion profiles of a HS{1,1} pulse with $\beta = 8/t_p$ and $\Delta f = 110$ MHz and a pulse length of (a) 100 ns, (b) 200 ns and (c) 400 ns. A 400 ns calculated pump excitation profiles next to the observer pulse excitation profiles is shown in (d).

It is expected that steeper excitation flanks lead to a smaller overlap with the observer pulses and therefore a smaller effect on the echo intensity. Despite this is the case here as well (Fig. 14d), the overlap is not reduced completely and 400 ns pulse still has some remaining spectral overlap with the observer pulses. We assume that the contradictive findings concerning the echo intensity here are caused by this remaining small overlap. It could become more perturbing for longer pulses as the overall energy of the pulses increases with the pulse length and therefore potential disturbances might be enhanced. A measurement with an larger offset between the pulses at 130 MHz shows that the echo decrease is indeed reduced (Fig. S13b) when the overlap gets smaller. Despite leading to a higher echo intensity, such a high offset is not favourable for nitroxode-nitroxide DEER, because of the limited width of the nitroxide spectrum.

4) The concentration dependence of the behavior is only discussed in a rather trivial and non-quantitative manner, despite the fact that it showed to be the major parameter influencing the improvement by the broadband pulses (comparison Figure6 and 7). That lower concentrations of spins are advantageous, especially for larger distances or broader distance distributions is well known in the community. Because broadband pulses might be especially interesting for these kind of systems, this should be discussed more quantitatively! The discussion in the SI including Figure S13 and the text after it is only very qualitative and rather trivial.

We have replaced the discussion in the SI with this section

[Figure]

**Figure S18:** The MNR-ratio of adiabatic and rectangular pulses as a function of the background density (with rectangular pulses) and the $\tau_{\text{truncation}}$-time. The corresponding maximum distance according to equation (13) of the main text is also depicted. As the background density reflects the concentration the x-axis is a measure for the concentration of the spin centres. In our sample with 80 µM, we had a background density $k_{\text{R}}$ of 0.1 with rectangular pulses.

Figure S18 shows that the performance of shaped pulses can heavily depend on the circumstances of the measurement. For a maximum distance below 4 nm ($\tau_{\text{truncation}} \approx 5$ µs), a MNR increase can be expected for all realistic concentration ranges. This is not the case if a longer distance shall be detected. For maximum distances around 5 nm, the MNR increase goes to 1 for high background densities of $k_{\text{R}} = 0.15$ 1/µs, which corresponds to very high concentrations > 100 µM. Typical concentrations for DEER measurements are around 50 µM, which here corresponds to a $k_{\text{R}} \approx 0.06$ 1/µs. For this concentration, a significant increase in the MNR can only be expected up to a truncation time of $\tau_{\text{truncation}} = 10$ µs, which is equal to a maximum distance of approximately 6 nm.

As broadband shaped pulses are particularly interesting for long distances, the calculations were performed up to a rather long truncation time of 25 µs (maximum distance of approximately 7.5 nm). For distances in the range > 6 nm, only with concentrations in the range of 10-30 µM ($k_{\text{R}} \approx 0.01$-0.04 1/µs) a significant increase in the MNR due to broadband shaped pump pulses can be expected. The MNR increase drops quickly when higher concentrations are used. For a maximum distance of 7.5 nm and for concentrations over approximately 40 µM no increase can be expected any more due to broadband shaped pulses. If a concentration of 80 µM is used, the MNR is about to decrease to roughly 40 % when switching to broadband shaped pulses. It is known that diluting the sample is favourable if long distances shall be detected because it increases the phase memory time of the echo (Schmidt et al., 2016). When broadband shaped pump pulses, the higher background decay

adds an additional point for carefully choosing the concentration of the sample and it seems to be advisable to avoid high concentrations.

**5) Minor point: In the supporting information equation (2) is wrong. After that there is a spelling error (ration).**

We thank Thomas Prisner for this remark. We have corrected the spelling mistakes.

Dear Reviewer,

We thank you much for your effort reading our manuscript so carefully and for your valuable advices. We strongly believe that the suggested changes and additions improved the manuscript a lot. Please find below a point-by-point reply to all your recommendations (in blue).

**[1] In section 3.3 (background decay), examine in detail how the type of shaped pulse affects the MNR if the recently published superior background correction method (kernel inclusion) is used. It is crucial to include this in this work.**

We agree with the reviewer that the novel approach including the background into the Kernel is a very efficient and convincing approach. We have added a new section in the main text where we clarified why we think that the MNR as we used it is the most feasible parameter for optimising settings for DEER.

As stated by equation (10) the measured raw data does not only consist of the desired form factor but includes a background contribution emerging from intermolecular interactions. A common way to deal with this, is to fit the background according to equation (11) and divide the raw data by the fit to obtain the form factor that can then be transformed into a distance distribution (Jeschke, 2012; Jeschke et al., 2006). When measuring DEER traces, a precise distance determination is desired. Since for an experimental parameter optimisation, the true underlying distance distribution is unknown, a metric is needed that is based on the recorded data. The MNR of the form factor is a suitable for this case as it increases with an increasing modulation depth and an increasing echo intensity. As the noise of the form factor increases towards its end due to the division by the background, the MNR goes down with a stronger background decay. It can therefore capture the fact that a larger background decay leads to less reliable distance distributions as has recently be investigated by [Fabregas, et. Al., 2020] in a detailed study. In their paper they also suggest a different method for background correction that treats the background by directly including it in the kernel that is needed to calculate the distance distribution from the DEER trace. As this methods renders the calculation of a form factor redundant, a MNR cannot be directly obtained by it. Even though this new method has shown itself to give more reliable distance distributions in the case of large background decays its performance still drops with an increasing background. Therefore, we consider the MNR that is obtained by the background correction by division still as the best measure to optimise settings for a DEER measurements experimentally.

We have also added a chapter in the SI to discuss the suitability of the MNR as a merit function if the background correction by kernel inclusion is used:

**S2 The MNR as the function of merit**

Here, we want to discuss whether the MNR is a suitable function of merit for the determination of distance distributions and up to which time point in the DEER trace, the MNR needs to be evaluated to serve this purpose. Therefore, we performed simulations with a model distance distribution $p_0$ that is based on the narrow distance distribution of the model system used in this study. We approximated the experimentally obtained distance distribution with a Gaussian with a mean at 5.08 nm and a standard deviation of 0.08 nm. We varied the background density in ten steps from $k = 0.01$ 1/µs to $k = 0.3$ 1/µs in combination with a low, medium and high noise level (noise $\sigma_0 = 0.02$, 0.05 and 0.1) that was added to the DEER trace. The background dimension was set to $d = 3$ and a modulation depth of 0.5 was used. The DEER traces were simulated in the time domain up to 8 µs. For each parameter set we generated ten different traces. To compare the background correction by division (Jeschke et al., 2006) with the kernel inclusion approach as described in (Fábregas Ibáñez and Jeschke, 2020) we analysed all simulated DEER traces with both methods. We did not fit the background but used the true background function. The regularisation parameter was chosen according to the generalised cross-validation method. The quality of the resulting distance distributions $p$ was estimated by the Euclidian distance $D$ from the true distance $p_0$:

$$D(p, p_0) = \|p - p_0\|_2 \tag{1}$$

The MNR of the form factor $F$ was calculated as described in the main text up to a limit of 7 µs according to equation (13) of the main text.

[Figure]

**Figure S2:** The Euclidian distance $D$ of the real and calculated distance distribution as defined in equation (1) is plotted as a function of the MNR. Each dot represents a simulated DEER trace with either low ($\sigma_0 = 0.02$, green), medium ($\sigma_0 = 0.05$, blue) and high ($\sigma_0 = 0.1$, red) noise. The background correction was performed by (a) dividing the DEER trace by the background and (b) including the background in the kernel.

In Fig. S2, the quality of the determined distance distribution was plotted as a function of the determined MNR for both a background correction by division (Fig. S2a) and a kernel inclusion approach (Fig. S2b). For each noise level the MNR only depends on the density of the background as all other parameters are kept constant and only the background density is varied. So a lower MNR corresponds to a higher background density rate and vice versa. For the low noise level ($\sigma_0 = 0.02$), the quality of the determined distance distributions only varies a little for different background density rates. For medium ($\sigma_0 = 0.05$) and high ($\sigma_0 = 0.1$) noise levels, however, the dependency of the quality of the determined distance distribution decreases significantly with a decreasing MNR. If the MNR is only evaluated up to an early point of the form factor, the

information of the background decay rate is lost in this case and is not properly included in the MNR as the MNR would then depend nearly exclusively on the given noise level.

[Figure]

**Figure S3:** An exemplary distance distribution obtained for a medium noise level ($\sigma_0 = 0.02$) with (a) a low background density ($k = 0.01$ MHz) and (b) a high background density ($k = 0.3$ MHz). The grey area shows the area that is covered by the calculated distance distribution for ten exemplary DEER traces. The mean of the shaded area is drawn in blue and the true distance is drawn in green.

A closer inspection reveals that whereas the obtained distance distributions for high background densities reproduce the mean of the distance distribution correctly, they overestimate the width of the distribution and the distance appears to be broader as it is (see Fig. S3 for an exemplary data set). Depending on the information that shall be obtained by the DEER measurements, the mean of the distance distribution might suffice. However, if high resolution distance distributions shall be obtained, it seems to be important to optimise the MNR up to the limit which is given by equation (13) of the main text. The comparison of both background correction methods shows that the kernel inclusion gives better results particularly for a high noise and a high background decay. It should therefore be considered as the superior method. However, the correlation between the quality of the determined distance distribution and the MNR is still valid. This is why, we consider the MNR as a proper function of merit, even if the kernel inclusion approach is used.

For a more comprehensive study, the effect of the MNR on the quality of the obtained distance distribution could also be tested for distance distributions with different distance ranges and widths. Such a detailed study was, however, beyond the scope of the this manuscript.

[2] Discuss in more detail whether and how the findings in this work are applicable to other samples (different distance distributions, different concentrations) and spectrometers (different resonator profiles, different Tx fidelity). From the current manuscript, itis unclear whether the findings are generalizable. This is important,

since it appears to be the purpose of the manuscript to make some general statements about experimental settings in DEER.

We have added a section in the main text to discuss the effect of different resonator profiles

[revised manuscript text omitted]
=0 at the center of the pulse.- Eq.(11): $(k*|t|)^{\wedge}(d/3)$ instead of $k*t^{\wedge}(d/3)$ - Eq.(12): A factor of 2 might be missing. -8.13: "i.e. a chirp pulse" - 12.18: Here, it is not clear how the numbers for the minimum detectable distance limit are obtained. - Kupce needs a grave accent on the c. Bohlen needs an umlaut on the o. - SI Eq.(2): t_truncation instead of tau_2

We thank the referee for these remarks and corrected and clarified these points.

**3. List of major changes**

We have made the following major changes in the manuscript. For clarity reasons we have not included the full sections here. They are marked as changes in the revised manuscript which starts on the next page.

1. In the introduction we have clarified the objective of this manuscript.
2. We have exchanged Fig.1 because we noticed a minor error in a sugar moiety of the tetravalent ligand.
3. A section where we discuss the suitability of the MNR as the function of merit, particularly when different background techniques are used, i.e. including the background in the kernel was added.
4. We added a section where we have discussed the pulse performance for different resonators based on simulated inversion profiles.
5. In the section "Background behaviour" and "Concentration dependence" we have added two section with a more quantitative discussion.

In the SI we have made the following changes:

1. The suitability of the MNR as merit function is discussed in a new section S2.
2. A new section S11 was added to discuss discrepancies from experimental and simulated inversion profiles of the broadband shaped pulses.
3. The former section S12 and S13 have been replaced by a new section S13 where we discuss the influence of the length of the broadband shaped pump pulses in more detail. Therefore, we have added experimentally recorded pulse inversion profiles for 100 ns, 200 ns and 400 ns broadband shaped pulses.
4. The full DEER traces of the sample with the high and low concentration are shown in anew section S20.
5. Major changes were made in the new section S21 to add a quantitative discussion of different concentration and distance ranges for the performance of broadband shaped pump pulses.

[revised manuscript text omitted]

**S1.1 EDFS**

The echo-detected-field sweep spectra were recorded with a Hahn echo sequence ( $\frac{\pi}{2}$ - $\tau$ - $\pi$ - $\tau$ − echo) pulse-sequence with $\tau = 1.5$ µs, a sweep width of 200 G and 10 shots per point, 3 scans and rectangular pulses. The length of the $\pi$-pulse was 16 ns at a frequency of 34 GHz.

**S1.2 Nutation experiments**

Pulse lengths of rectangular and Gaussian pulses were determined with nutation experiments with the pulse sequence (inversion pulse $- \tau_1 - \frac{\pi}{2} - \tau_2 - \pi - \tau_2 -$ echo). $\tau_1$ was set to 1 μs.

**S1.3 Resonator profile**

The resonator profile was measured by a series of nutation experiments at different frequencies as described in the literature (Doll and Jeschke, 2014). It was measured over 300 MHz with a step size of 10 MHz. The magnetic field was co-stepped. The nutation frequencies were calculated by a Fourier transformation of the nutation traces.

**S1.4 DEER**

All DEER experiments were measured with the standard four pulse DEER sequence:

$$\frac{\pi}{2}_{obs} - \tau_1 - \pi_{\mathrm{obs}} - t - \pi_{\mathrm{pump}} - \tau_1 + \tau_2 - t - \pi_{\mathrm{obs}} - \tau_2 - \mathrm{echo}$$

The delay between the $\pi/2$ and the $\pi$ pulse in the observer channel $\tau_1$ was 400 ns. The dipolar evolution time $\tau_2$ was 8 μs. For all DEER experiments with rectangular and Gaussian pump pulses the pump frequency was set to 34.00 GHz. The magnetic field was chosen such that the pump lies on the maximum of the nitroxide spectrum. We used the phase cycling ((x) [x] $x_p$ x) as suggested by (Tait and Stoll, 2016) and nuclear modulation averaging as suggested by (Jeschke, 2012).

**S1.5 DEER optimisation**

For the optimisation measurements we used a python script that can automatically perform several DEER experiments after another. We shifted the magnetic field from 1.2090 T to 1.2113 T for an observer pulse of 33.91 GHz and from 1.2097 T to 1.2119 T for an observer position of 33.93 GHz to ensure that the pump pulse is on the maximum of the nitroxide spectrum. Figure S1 illustrates the idea with a fixed observer frequency of 33.93 GHz.

[Figure]

**Figure S1:** The resonator profile (green dots) with the different offsets during an optimisation measurement. The nitroxide spectrum (orange) is shifted with the offset. The observer frequency stays fixed at 33.93 GHz and is indicated by a blue line. The shift of the pump spin is indicated by the red line.

**S1.6 Pulse calculations**

For rectangular and Gaussian pulses, we used the pulses that are generated by Bruker Xepr software. For Gaussian pulses the FWHM is defined by FWHM $= \frac{t_{\text{p}}}{2\sqrt{2\ln(2)}}$. All other pulses were calculated with the *pulse* function from the *easyspin* (Version 5.2.21) package for MATLAB R2018b (Stoll and Schweiger, 2006). The resulting pulse shapes were normalised to amplitude values between -1 and 1 and loaded into Xepr.

**S1.7 Integration window**

The integration window was determined by recording a series of 300 Hahn echoes in transient mode. We evaluated the signal-to-noise (SNR) with different integration windows and determined the integration window with the maximum SNR.

**S1.8 Inversion profiles**

Inversion profiles for broadband shaped pulses were measured with the pulse sequence.

$$\text{broadband shaped pulse} - \tau_1 - \frac{\pi}{2}_{obs} - \tau_2 - \pi_{\text{obs}} - \tau_2 - \text{echo}$$

The inversion profiles were measured as the echo intensity as function of the frequency offset of the initial broadband shaped pulses. The $\pi/2$ and $\pi$ pulses were rectangular pulses with a fixed frequency of 34 GHz.

**S2S2 The MNR as the function of merit**

Here, we want to discuss whether the MNR is a suitable function of merit for the determination of distance distributions and up to which time point in the DEER trace, the MNR needs to be evaluated to serve this purpose. Therefore, we performed simulations with a model distance distribution $p_0$ that is based on the narrow distance distribution of the model system used in this study. We approximated the experimentally obtained distance distribution with a Gaussian with a mean at 5.08 nm and a standard deviation of 0.08 nm. We varied the background density in ten steps from $k = 0.01$ 1/µs to $k = 0.3$ 1/µs in combination with a low, medium and high noise level (noise $\sigma_0 = 0.02$, 0.05 and 0.1) that was added to the DEER trace. The background dimension was set to $d = 3$ and a modulation depth of 0.5 was used. The DEER traces were simulated in the time domain up to 8 µs. For each parameter set we generated ten different traces. To compare the background correction by division (Jeschke et al., 2006) with the kernel inclusion approach as described in (Fábregas Ibáñez and Jeschke, 2020) we analysed all simulated DEER traces with both methods. We did not fit the background but used the true background function. The regularisation parameter was chosen according to the generalised cross-validation method. The quality of the resulting distance distributions $p$ was estimated by the Euclidian distance $D$ from the true distance $p_0$:

$$D(p, p_0) = \|p - p_0\|_2 \tag{1}$$

The MNR of the form factor $F$ was calculated as described in the main text up to a limit of 7 µs according to equation (13) of the main text.

[Figure]

**Figure S2:** The Euclidian distance $D$ of the real and calculated distance distribution as defined in equation (1) is plotted as a function of the MNR. Each dot represents a simulated DEER trace with either low ($\sigma_0 = 0.02$, green), medium ($\sigma_0 = 0.05$, blue) and high ($\sigma_0 = 0.1$, red) noise. The background correction was performed by (a) dividing the DEER trace by the background and (b) including the background in the kernel.

In Fig. S2, the quality of the determined distance distribution was plotted as a function of the determined MNR for both a background correction by division (Fig. S2a) and a kernel inclusion approach (Fig. S2b). For each noise level the MNR only depends on the density of the background as all other parameters are kept constant and only the background density is varied. So a lower MNR corresponds to a higher background density rate and vice versa. For the low noise level ($\sigma_0 = 0.02$), the quality of the determined distance distributions only varies a little for different background density rates. For medium ($\sigma_0 = 0.05$) and high ($\sigma_0 = 0.1$) noise levels, however, the dependency of the quality of the determined distance distribution decreases significantly with a decreasing MNR. If the MNR is only evaluated up to an early point of the form factor, the information of the background decay rate is lost in this case and is not properly included in the MNR as the MNR would then depend nearly exclusively on the given noise level.

[Figure]

**Figure S3:** An exemplary distance distribution obtained for a medium noise level ($\sigma_0 = 0.02$) with (a) a low background density ($k = 0.01$ MHz) and (b) a high background density ($k = 0.3$ MHz). The grey area shows the area that is covered by the

calculated distance distribution for ten exemplary DEER traces. The mean of the shaded area is drawn in blue and the true distance is drawn in green.

A closer inspection reveals that whereas the obtained distance distributions for high background densities reproduce the mean of the distance distribution correctly, they overestimate the width of the distribution and the distance appears to be broader as it is (see Fig. S3 for an exemplary data set). Depending on the information that shall be obtained by the DEER measurements, the mean of the distance distribution might suffice. However, if high resolution distance distributions shall be obtained, it seems to be important to optimise the MNR up to the limit which is given by equation (13) of the main text. The comparison of both background correction methods shows that the kernel inclusion gives better results particularly for a high noise and a high background decay. It should therefore be considered as the superior method. However, the correlation between the quality of the determined distance distribution and the MNR is still valid. This is why, we consider the MNR as a proper function of merit, even if the kernel inclusion approach is used.

For a more comprehensive study, the effect of the MNR on the quality of the obtained distance distribution could also be tested for distance distributions with different distance ranges and widths. Such a detailed study was, however, beyond the scope of the this manuscript.

**S3 Determination of the integration window**

To determine the ideal integration window we recorded a series of Hahn echoes and calculated the SNR ratio for different integration window lengths. The results show that for rectangular pulses the ideal integration window is typically longer than the $\pi$-pulse length (Fig. S2S4). An improvement of up to 14 % for a $\pi$-pulse length of 28 ns and an ideal integration window of 44 ns was achieved.

[Figure]

**Figure S2S4:** The SNR for rectangular pulses of a series of transient Hahn echoes is shown as a function of the integration window length. The red lines indicate the integration window with the maximum SNR, the blue lines indicate an integration window that has the length of the $\pi$-pulse. The pulses have the following settings: a) frequency: 33.91 GHz, amplitude: 100 %. b) frequency: 33.91 GHz, amplitude: 60 %. c) frequency: 33.93 GHz, amplitude: 100 %. d) frequency: 33.93 GHz, amplitude: 60 %.

For Gaussian pulses the ideal integration window is typically smaller than the $\pi$-pulse length (figure S3Fig. S5).

[Figure]

**Figure S3S5:** The SNR for Gaussian pulses of a series of transient Hahn echoes is shown as a function of the integration window length. The red lines indicate the integration window with the maximum SNR, the blue lines indicate an integration window that has the length of the $\pi$-pulse. The pulses have the following settings: a) frequency: 33.91 GHz, amplitude: 100 %. b) frequency: 33.91 GHz, amplitude: 60 %. c) frequency: 33.93 GHz, amplitude: 100 %. d) frequency: 33.93 GHz, amplitude: 60 %.

**S3S4 Parameters for the observer pulse**

**Table S1:** Parameters for the rectangular observer pulses. The pulse length is referring to the $\pi$-pulse.

| $f_{obs}$ [GHz] | Obs. Amp. [%] | $t_\pi$ [ns] | Length of integration window [ns] |
|---|---|---|---|
| 33.91 | 100 | 28 | 44 |
| | 60 | 32 | 48 |
| 33.93 | 100 | 28 | 44 |
| | 60 | 32 | 48 |

**Table S2:** Parameters for the Gaussian observer pulses. The pulse length is referring to the $\pi$-pulse.

| $f_{obs}$ [GHz] | Obs. Amp. [%] | $t_\pi$ [ns] | Length of integration window [ns] |
|---|---|---|---|
| 33.91 | 100 | 56 | 48 |
| | 60 | 74 | 56 |
| 33.93 | 100 | 56 | 52 |
| | 60 | 74 | 56 |

**S4S5 The MNR for rectangular and Gaussian pump pulses evaluated up to 7 μs**

**Table S3:** MNR for a rectangular pump pulse and different rectangular observer pulses. The pump pulses had a length of 16 ns. The MNR has been evaluated up to 7 μs.

| $f_{obs}$ [GHz] | Obs. Amp. [%] | $t_\pi$ [ns] | MNR | Mod depth $\lambda$ |
|---|---|---|---|---|
| **33.91** | 100 | 28 | 30 | 0.32 |
| | 60 | 32 | 32 | 0.32 |
| **33.93** | 100 | 28 | 32 | 0.31 |
| | 60 | 32 | 35 | 0.31 |

**Table S4:** MNR for a Gaussian pump pulse and different Gaussian observer pulses. The pump pulses had a length of 34 ns. The MNR has been evaluated up to 7 μs.

| $f_{obs}$ [GHz] | Obs. Amp. [%] | $t_\pi$ [ns] | MNR | Mod depth $\lambda$ |
|---|---|---|---|---|
| **33.91** | 100 | 56 | 36 | 0.31 |
| | 60 | 74 | 32 | 0.29 |
| **33.93** | 100 | 56 | 41 | 0.31 |
| | 60 | 74 | 38 | 0.31 |

**S6 Inversion profiles for rectangular and Gaussian pulses**

[Figure]

**Figure S6:** The excitation profiles of the observer (blue) and pump pulses (red) of (a) rectangular and (b) Gaussian pulses. The light blue profiles are for the $\pi/2$ observer pulses and the dark blue profiles for the $\pi$ observer pulse. The rectangular observer pulses have an amplitude of 60 %, a pulse length of 32 ns ($\pi$ on observer) and 16 ns (pump), and the Gaussian have an amplitude of 100 %, a length of 56 ns ($\pi$ on observer) and 34 ns (pump). It can be seen that the spectral overlap can be reduced with Gaussian pulses. The pulse amplitudes of the pump pulse are always 100 %.

**S6S7 Simulations of spin inversion trajectories**

We simulated the effect of an HS{1,1} pulse with a pulse length of 100 ns, a truncation parameter of $\beta = 8/t_p$, a frequency sweep range from -55 MHz and 55 MHz. We performed the numerical simulation in the density operator framework with MATLAB R2018b. The maximum of the $B_1$-field was set to 30 MHz, which corresponds to the maximum of the resonator profile. This pulse shows a good inversion between a frequency range of approximately -40 MHz and 40 MHz (figureFig. S7a). In figureFig. S7b, the inversion of a spin packet with an offset of -40 MHz and 40 MHz are shown. It can be seen that the spins are inverted in a time window between roughly 20 ns and 80 ns, making an effective pulse length of 60 ns. This correspondswould correspond to a minimum distance of 2.32 nm. Note that these numbers were only obtained by visual inspection, so this should only be considered as a qualitative discussion. The spin flip behaviours is also different for different pulses.

[Figure]

**Figure S5S7:** a) The excitation profile of HS{1,1} with a pulse length of 100 ns, a truncation of $\beta = 8/t_p$ and a frequency range from -55 MHz and +55 MHz. b) The inversion of a spin packet with an offset of -40 MHz (blue) and +40 MHz (green).

**S7S8 The MNR for broadband pump pulses evaluated up to 7 μs**

**Table S5:** MNR for the different broadband shaped pulses with a rectangular observer pulse. The MNR has been evaluated up to 7 μs.

| $f_{obs}$ [GHz] | Obs. Amp. [%] | Pump pulse | $t_\pi$ [ns] | $\Delta f$ [MHz] | Offset [MHz] | MNR | Mod. depth $\lambda$ |
|---|---|---|---|---|---|---|---|
| 33.91 | 100 | HS{1,6} ($\beta = 8/t_p$) | 100 | 110 | 90 | 32 | 0.63 |
| | | WURST ($n$=6) | 100 | 160 | 90 | 37 | 0.64 |
| | | Chirp ($t_r = 30$ ns) | 36 | 120 | 90 | 33 | 0.50 |
| | | HS{1,1} ($\beta = 6/t_p$) | 100 | 110 | 90 | 41 | 0.57 |
| | 60 | HS{1,6} ($\beta = 10/t_p$) | 100 | 110 | 90 | 31 | 0.62 |
| | | WURST ($n$=6) | 100 | 160 | 90 | 30 | 0.64 |
| | | Chirp ($t_r = 30$ ns) | 100 | 160 | 90 | 32 | 0.67 |
| | | HS{1,1} ($\beta = 6/t_p$) | 100 | 110 | 90 | 34 | 0.57 |
| 33.93 | 100 | HS{1,6} ($\beta = 10/t_p$) | 100 | 110 | 100 | 39 | 0.57 |
| | | WURST ($n$=6) | 100 | 160 | 100 | 39 | 0.62 |
| | | Chirp ($t_r = 10$ ns) | 36 | 120 | 90 | 38 | 0.47 |
| | | HS{1,1} ($\beta = 6/t_p$) | 100 | 110 | 90 | 43 | 0.53 |
| | 60 | HS{1,6} ($\beta = 10/t_p$) | 100 | 110 | 90 | 40 | 0.60 |
| | | WURST ($n$=6) | 100 | 120 | 100 | 36 | 0.57 |
| | | Chirp ($t_r = 10$ ns) | 36 | 120 | 90 | 38 | 0.47 |
| | | HS{1,1} ($\beta = 8/t_p$) | 100 | 90 | 80 | 43 | 0.48 |

**Table S6:** MNR for the different broadband shaped pulses with a Gaussian observer pulse. The MNR has been evaluated up to 7 µs.

| $f_{obs}$ [GHz] | Obs. Amp. [%] | Pump pulse | $t_\pi$ [ns] | $\Delta f$ [MHz] | Offset [MHz] | MNR | Mod. depth $\lambda$ |
|---|---|---|---|---|---|---|---|
| 33.91 | 100 | HS{1,6} ($\beta = 10/t_p$) | 100 | 90 | 90 | 36 | 0.60 |
| | | WURST ($n$=6) | 100 | 160 | 90 | 30 | 0.64 |
| | | Chirp ($t_r = 10$ ns) | 36 | 120 | 80 | 37 | 0.50 |
| | | HS{1,1} ($\beta = 6/t_p$) | 100 | 110 | 90 | 40 | 0.58 |
| | 60 | HS{1,6} ($\beta = 8/t_p$) | 100 | 110 | 90 | 38 | 0.63 |
| | | WURST ($n$=6) | 100 | 160 | 100 | 34 | 0.48 |
| | | Chirp ($t_r = 10$ ns) | 100 | 120 | 90 | 38 | 0.48 |
| | | HS{1,1} ($\beta = 6/t_p$) | 100 | 110 | 90 | 37 | 0.58 |
| 33.93 | 100 | HS{1,6} ($\beta = 10/t_p$) | 100 | 90 | 90 | 39 | 0.57 |
| | | WURST ($n$=6) | 100 | 160 | 100 | 38 | 0.62 |
| | | Chirp (no smoothing) | 36 | 120 | 80 | 45 | 0.49 |
| | | HS{1,1} ($\beta = 8/t_p$) | 100 | 110 | 90 | 50 | 0.52 |
| | 60 | HS{1,6} ($\beta = 10/t_p$) | 100 | 110 | 90 | 45 | 0.61 |
| | | WURST ($n$=6) | 100 | 160 | 90 | 40 | 0.63 |
| | | Chirp ($t_r = 9$ ns) | 36 | 120 | 80 | 45 | 0.47 |
| | | HS{1,6} ($\beta = 8/t_p$) | 100 | 110 | 80 | 47 | 0.52 |

[Figure]

**Figure S6S8:** The excitation profiles of the best performing (a) HS{1,6}, (b) WURST, (c) chirp and (d) HS{1,1} pulse. The parameters of the pump and observer pulses can be found in table 3 of the main text.

[Figure]

**Figure S7S9:** The pulse shapedshapes of anthe best performing (a) HS{1,6} , (b) WURST , (c) chirp  and (d) HS{1,1}  pulse with the real part (green) and imaginary part (blue). The parameters broadband shaped pulses can be found in table 23 of the main text.

[Figure]

**Figure S10:** The simulated (green) and experimentally recorded (blue) inversion profiles of the best performing (a) HS{1,6}, (b) WURST, (c) chirp and (d) HS{1,1} pulse. The parameters of the pump and observer pulses can be found in table 3 of the main text.

We recorded the inversion profiles of the best performing pulses and compared them with the simulations in order to detect potential deviations. The results in Fit. S10 shows that the experimentally recorded inversion profiles reproduce the general trends of the simulations. Nonetheless, there are some deviations that are probably caused by the spectromeer and that shall be discussed here.

It can be noticed that for HS{1,6} and HS{1,1} pulses the measured inversion profiles do not reach the inversion profile of the simulation. For HS{1,6} the overall inversion efficiency is a bit lower than expected and for HS{1,1} pulses a bump in the centre of the frequency sweep was found. The simulations can, however, predict the fact that HS{1,6} have a higher inversion efficiency than HS{1,1} pulses. For WURST and chirp pulses the experimentally recorded inversion profiles reach the inversion efficiency of the simulation.

The experimental inversion profiles of HS{1,6}, HS{1,1} and chirp pulses have a larger inversion range than what is predicted by the simlations. This can increase the overlap with the observer pulse and therefore reduce the echo intensity. But as the inversion range is only a little bit larger, we consider this not to be particularly worrysome. For HS{1,6} and HS{1,1} pulses, the larger inversion range could compensate the reduced inversion efficiency

For the chirp pulses the frequency range as well as the inversion efficiency of the experimental and simulated inversion profiles agree. There are some minor deviations in the pattern of the oscillations that are present in the inversion profile, which we do not expect to have a large effect on the performance of the pulse.

**S12 Full** DEER traces

[Figure]

**Figure S8S11:** Comparison of the performance of DEER with rectangular pulses (green) and with Gaussian observer pulses and the HS{1,1} pulse from table 1 that yielded the best MNR (blue). The form factors are shown in (a) and the corresponding distance distributions in (b). One 10 minute scan was recorded for both experiments. The corresponding DEER traces are depicted in Fig. S17.

**S13 S11****The influence of the length of broadband shaped pump pulses**

Tests with broadband shaped pump pulses with pulse lengths of 200 ns and 400 ns showed that they do not lead to an overall performance increase. This is shown here exemplary by comparing the performance of HS{1,1} pump pulses and Gaussian observer pulses (Fig. S12). There are indeed some pump pulses (for example a HS{1,1} pulse with $\beta = 10/t_p$ and $\Delta f = 110$ MHz) that show an improvement with a longer pulse length, however there is no overall gain by using a pump pulse length of 200 ns.

[Figure]

**Figure S12:** HS{1,1} pump pulses of (a) 100 ns and (b) 200 ns length. The observer pulses were Gaussian pulses with 100 % intensity at an observer position with a 90 MHz offset from the centre of the resonator profile and a pulse length of 56 ns for the $\pi$ pulse. The colour bars are normalised to the same value so that both heat maps are comparable.

We noticed that a major problem with longer broadband shaped pump pulses is that the intensity of the echo can be reduced (Fig. S13a). For a pump pulse offset of 90 MHz, the echo intensity at the zero time of the DEER trace is reduced significantly when increasing the pump pulse lengths from 100 ns over 200 ns to 400 ns.

[Figure]

A comparison of the calculated inversion profiles of the respective pulses (Fig. S14a-c) shows that, whereas the 100 ns pulse should lead to an incomplete inversion, a nearly complete inversion can be expected for the longer pulses. Furthermore, the longer pulses should have slightly steeper excitation flanks. Those trends can indeed be found for the measured inversion profiles. There are some deviations of the measured and calculated inversion profiles. The measured inversion profile of the 100 ns pulse shows an increased frequency width compared to the calculated profile. Furthermore, there is bump in the centre of the frequency sweep. The measured inversion profiles of the longer pulses show the expected steep frequency flanks that can also be seen in the simulation. The inversion profiles of the 200 ns and the 400 ns pulses show a small asymmetry around the centre of the frequency sweep. We assign these deviations to instrumental pulse distortions caused by the spectrometer.

[Figure]

**Figure S14:** Calculated and measured inversion profiles of a HS{1,1} pulse with $\beta = 8/t_p$ and $\Delta f = 110$ MHz and a pulse length of (a) 100 ns, (b) 200 ns and (c) 400 ns. A 400 ns calculated pump excitation profiles next to the observer pulse excitation profiles is shown in (d).

It is expected that steeper excitation flanks lead to a smaller overlap with the observer pulses and therefore a smaller effect on the echo intensity. Despite this is the case here as well (Fig. 14d), the overlap is not reduced completely and 400 ns pulse still

has some remaining spectral overlap with the observer pulses. We assume that the contradictive findings concerning the echo intensity here are caused by this remaining small overlap. It could become more perturbing for longer pulses as the overall energy of the pulses increases with the pulse length and therefore potential disturbances might be enhanced. A measurement with an larger offset between the pulses at 130 MHz shows that the echo decrease is indeed reduced (Fig. S13b) when the overlap gets smaller. Despite leading to a higher echo intensity, such a high offset is not favourable for nitroxode-nitroxide DEER, because of the limited width of the nitroxide spectrum.

**S14 The influence of the $B_1$ field strength on chirp pulses**

[Figure]

**Figure S15:** The inversion profiles of different chirp pulses with (a) a pulse length of 36 ns and no quarter sine smoothing, (b) a pulse length of 36 ns and a quarter sine smoothing with $t_r = 10$ ns and (c) a pulse length of 100 ns and a quarter sine smoothing with $t_r = 10$ ns. The frequency width of all pulses is $\Delta f = 120$ MHz.

**S15 Comparison of bandwidth compensated and non-bandwidth compensated pulses**

We tested the performance of a bandwidth compensation for HS{1,6}, WURST, chirp and HS{1,1} pump pulses. The observer pulses were rectangular with an offset of 90 MHz and an observer $\pi$ pulse length of 28 ns. We estimated the effect of bandwidth

compensation with the help of the $\eta_{2\text{p}}$ parameter. For WURST and chirp pulses, a bandwidth compensation lead to an improvement of 3.0 % and 3.2 %. However, for HS{1,6} and HS{1,1} pulses, we observed a decrease of 10.5% and 2.6% (data not shown). As a bandwidth compensation requires a measurement of the resonator profile before each DEER measurement and did not always result in an increase in performance, we decided to stick to pulses without bandwidth compensation.

**S12 Comparison of 100 ns and 200 ns pulse lengths**

[Figure]

**Figure S9:** HS{1,1} pump pulses of (a) 100 ns and (b) 200 ns length. The observer pulses were rectangular with 100 % intensity at an observer position of 33.91 GHz and a pulse length of 28 ns for the $\pi$ pulse.**S16**

**S13 Echo decrease with long broadband shaped pump pulses**

We have checked the echo intensity at the zero time of the DEER trace with different pump pulses. There is always a slight decrease of the echo intensity in the presence of a pump pulse (Fig. S8a). It is, however, negligible for rectangular pump pulses. With a 100 ns HS{1,1} pulse there is a stronger decrease of the echo, which gets even worse for 200 ns and 400 ns pulses. In the last case, nearly the whole echo has disappeared. The reason for this behaviour is not entirely clear to us as the excitation profiles do not change significantly with the pulse length (Fig. S8 b). However, we chose not to investigate 200 ns and 400 ns pulses.

[Figure]

**Figure S10:** a) The echoes are at the zero time of the DEER trace. The echoes are recorded without a pump pulse (red), with a rectangular pump pulse (blue), with a 100 ns (green), 200 ns (yellow) and 400 ns (purple) HS{1,1} pump pulse (b) shows the excitation profile of the respective HS{1,1} pump pulses. Their parameters are $\beta = 6/t_p$, $\Delta f = 90$ MHz, offset to observer: 80 MHz and a pulse length of 100 ns (green), 200 ns (yellow) and 400 ns (purple).

**S14 The MNR for rectangular and Gaussian pump pulses evaluated up to 2 µs**

**Table S7:** MNR for a rectangular pump pulse and different rectangular observer pulses. The pump pulses had a length of 16 ns. The MNR has been evaluated up to 2 µs.

| $f_{obs}$ [GHz] | Obs. Amp. [%] | $t_\pi$ [ns] | MNR | Mod. depth $\lambda$ |
|---|---|---|---|---|
| 33.91 | 100 | 28 | 41 | 0.32 |
|  | 60 | 32 | 41 | 0.32 |
| 33.93 | 100 | 28 | 40 | 0.31 |
|  | 60 | 32 | 44 | 0.31 |

**Table S8:** MNR for a Gaussian pump pulse and different Gaussian observer pulses. The pump pulses had a length of 34 ns. The MNR has been evaluated up to 2 µs.

| $f_{obs}$ [GHz] | Obs. Amp. [%] | $t_\pi$ [ns] | MNR | Mod. depth $\lambda$ |
|---|---|---|---|---|

| 33.91 | 100 | 56 | 50 | 0.31 |
|-------|-----|----|----|------|
|       | 60  | 74 | 42 | 0.29 |
| 33.93 | 100 | 56 | 53 | 0.31 |
|       | 60  | 74 | 48 | 0.31 |

**S15S17 The MNR for broadband pump pulses evaluated up to 2 μs**

**Table S9:** MNR for the different broadband shaped pulses with a rectangular observer pulse. The MNR has been evaluated up to 2 μs.

| $f_{obs}$ [GHz] | Obs. Amp. [%] | Pump pulse | $t_\pi$ [ns] | $\Delta f$ [MHz] | Offset [MHz] | MNR | Mod. depth $\lambda$ |
|---|---|---|---|---|---|---|---|
| 33.91 | 100 | HS{1,6} ($\beta = 8/t_p$) | 100 | 110 | 90 | 58 | 0.63 |
| | | WURST ($n$=6) | 100 | 160 | 90 | 60 | 0.64 |
| | | Chirp ($t_r = 30$ ns) | 36 | 120 | 90 | 52 | 0.50 |
| | | HS{1,1} ($\beta = 6/t_p$) | 100 | 110 | 90 | 64 | 0.57 |
| | 60 | HS{1,6} ($\beta = 10/t_p$) | 100 | 110 | 90 | 59 | 0.62 |
| | | WURST ($n$=6) | 100 | 160 | 90 | 63 | 0.63 |
| | | Chirp ($t_r = 30$ ns) | 100 | 160 | 90 | 57 | 0.66 |
| | | HS{1,1} ($\beta = 6/t_p$) | 100 | 110 | 90 | 53 | 0.57 |
| 33.93 | 100 | HS{1,6} ($\beta = 10/t_p$) | 100 | 110 | 100 | 60 | 0.57 |
| | | WURST ($n$=6) | 100 | 160 | 100 | 63 | 0.62 |
| | | Chirp ($t_r = 10$ ns) | 36 | 120 | 90 | 57 | 0.47 |
| | | HS{1,1} ($\beta = 6/t_p$) | 100 | 110 | 90 | 65 | 0.53 |
| | 60 | HS{1,6} ($\beta = 10/t_p$) | 100 | 110 | 90 | 64 | 0.60 |
| | | WURST ($n$=6) | 100 | 120 | 100 | 59 | 0.57 |
| | | Chirp ($t_r = 10$ ns) | 36 | 120 | 90 | 58 | 0.47 |
| | | HS{1,1} ($\beta = 8/t_p$) | 100 | 90 | 80 | 58 | 0.48 |

**Table S10:** MNR for the different broadband shaped pulses with a Gaussian observer pulse. The MNR has been evaluated up to 2 µs. The chirp pulse where no $t_r$ time is specified is a pulse without the quartersine smoothing.

| $f_{obs}$ [GHz] | Obs. Amp. [%] | Pump pulse | $t_\pi$ [ns] | $\Delta f$ [MHz] | Offset [MHz] | MNR | Mod. depth $\lambda$ |
|---|---|---|---|---|---|---|---|
| 33.91 | 100 | HS{1,6} ($\beta = 10/t_p$) | 100 | 90 | 90 | 65 | 0.60 |
| | | WURST ($n=6$) | 100 | 160 | 90 | 64 | 0.64 |
| | | Chirp ($t_r = 10$ ns) | 36 | 120 | 80 | 59 | 0.50 |
| | | HS{1,1} ($\beta = 6/t_p$) | 100 | 110 | 90 | 68 | 0.58 |
| | 60 | HS{1,6} ($\beta = 8/t_p$) | 100 | 110 | 90 | 71 | 0.63 |
| | | WURST ($n=6$) | 100 | 160 | 100 | 67 | 0.63 |
| | | Chirp ($t_r = 10$ ns) | 100 | 120 | 90 | 59 | 0.48 |
| | | HS{1,1} ($\beta = 6/t_p$) | 100 | 110 | 90 | 63 | 0.58 |
| 33.93 | 100 | HS{1,6} ($\beta = 10/t_p$) | 100 | 90 | 90 | 73 | 0.57 |
| | | WURST ($n=6$) | 100 | 160 | 100 | 72 | 0.62 |
| | | Chirp (no smoothing) | 36 | 120 | 80 | 65 | 0.49 |
| | | HS{1,1} ($\beta = 8/t_p$) | 100 | 110 | 90 | 71 | 0.52 |
| | 60 | HS{1,6} ($\beta = 10/t_p$) | 100 | 110 | 90 | 82 | 0.61 |
| | | WURST ($n=6$) | 100 | 160 | 90 | 73 | 0.63 |
| | | Chirp ($t_r = 9$ ns) | 36 | 120 | 80 | 63 | 0.47 |
| | | HS{1,1} ($\beta = 8/t_p$) | 100 | 110 | 80 | 74 | 0.52 |

**Table S11:** MNR for the different broadband shaped pulses with a Gaussian observer pulse. The MNR has been evaluated up to 2 µs. The chirp pulse where no $t_r$ time is specified is a pulse without the quartersine smoothing.

| $f_{obs}$ [GHz] | Obs. Amp. [%] | Pump pulse | $t_\pi$ [ns] | $\Delta f$ [MHz] | Offset [MHz] | MNR | Mod. depth $\lambda$ |
|---|---|---|---|---|---|---|---|
| 33.93 | 100 | HS{1,6} ($\beta = 10/t_p$) | 100 | 90 | 90 | 61 | 0.55 |
| | | WURST ($n$=6) | 100 | 160 | 100 | 54 | 0.59 |
| | | Chirp (no smoothing) | 36 | 120 | 80 | 58 | 0.46 |
| | | HS{1,1} ($\beta = 8/t_p$) | 100 | 110 | 90 | 65 | 0.47 |
| | 60 | HS{1,6} ($\beta = 10/t_p$) | 100 | 110 | 90 | 59 | 0.56 |
| | | WURST ($n$=6) | 100 | 160 | 90 | 53 | 0.58 |
| | | Chirp ($t_r = 9$ ns) | 36 | 120 | 80 | 54 | 0.43 |
| | | HS{1,1} ($\beta = 8/t_p$) | 100 | 110 | 80 | 55 | 0.46 |

5     **S17S19 Correlation between the background density and the modulation depth**

[Figure]

[Figure]

**Figure S11S16:** The correlation between the modulation depth and the background. Each dot represents a DEER trace that has been measured in the course of this study. Theoretically, the modulation depth and the background density should lie on a line through the origin. This is in fact roughly the case. The determination of the background density $k$ seems to give a rather large error, which causes the deviations from the expected result. The fitted line has a slope of 3.43 μs and an $x$-axis distance of -0.06.

**S18S20 Background decay of the DEER traces**

[Figure]

[Figure]

**Figure S17:** The (normalised) experimental raw data of the sample with a 80 µM (a) and 30 µM (b) ligand concentration. The settings for GG (green) were performed with a 100 % pulse amplitude and a 70 MHz offset. For GS (blue), the observer pulses were at a frequency of 70 MHz offset from the centre of the resonator. The pump pulses were HS{1,1} pulses, with the parameters $\beta = 8/t_p$, $t_p = 100$ ns, $\Delta f = 110$ MHz and an offset from the observer pulse of 90 MHz. Note that the acquisition time for the sample with lower concentration was longer in order to reach a similar noise level for both cases. The corresponding form factors are depicted in Fig. S11.

**S21 Calculation of the background-dependent performance of broadband shaped pulses**

To estimate the influence of  broadband shaped pulses for different maximum distances and concentrations, we performed some analytical calculations. The background decay reduces the echo-intensity and therefore decreases the signal-to-noise ratio towards the end of the DEER trace. Whereas the measured trace $V(t)$ has a constant noise level $\sigma_0$, the background corrected form factor has an increasing noise level towards the end:

$$\sigma(t) = \sigma_0 \exp(kt),$$

(2)

where $\sigma(t)$ is the noise of the form factor and $k$ is the background density. Here, we assumed a 3D background. As discussed in the main text, the form factor is truncated at a time $t_{\text{truncation}}$ to exclude the later part. An integration from $t = 0$ to $t = \tau_{\text{truncation}}$, with $\tau_{\text{truncation}}$ as the dipolar evolution time, yields the average noise in the form factor

$$\sqrt{<\sigma^2>} = \sigma_0 \sqrt{\frac{1}{2k\tau_{\text{truncation}}} \left( \exp(2k\tau_{\text{truncation}}) - 1 \right)}.$$

 (3)

The modulation-to-noise (MNR) as the ratio of the modulation depth $\lambda$ and the average noise is then described by:

$$\text{MNR} = \frac{\lambda}{\sigma_0}\sqrt{\frac{2k\tau_{\text{truncation}}}{\exp(2k\tau_{\text{truncation}})-1}}.$$

($\cancel{3}$4)

As both the modulation depth $\lambda$ and the background density $k$ directly depend on the inversion efficiency, a linear dependence can be expected between them. Indeed, we experimentally found an approximately linear correlation between them (Fig. $\cancel{\text{S11}}$S16). Whereas the $\eta_{2P}$ value captures a decrease in echo intensity it will miss the effect of a larger background decay. We chose exemplary parameters that resembled our experimental findings. For the modulation depth, we assumed an increase from 30 % to 50 $\cancel{\text{%.}}$%, which corresponds to the modulation depths that we found for rectangular and the best HS{1,1} pump pulse. For the density of the background we assumed an increase about the same factor: $k_S = \frac{5}{3}k_R$ with $k_S$ as the background density for the broadband shaped pulse and $k_R$ as the background density for the rectangular $\cancel{\text{pulse. According to equation (4}}$pulses. As the sample with a concentration of 80 µM of doubly-labelled ligand had a background density $k_R$ of approximately 0.1 with rectangular pulses, we tested $k_R$ –values from 0 to 0.15 1/µs to keep it in a realistic range. According to equation (5) this will give an MNR increase of

$$\text{MNR}_{\text{increase}} = \frac{5}{3}\sqrt{\frac{5\left(\exp(2k_R\tau_{\text{truncation}})-1\right)}{3\left(\exp\left(\frac{10}{3}k_R\tau_{\text{truncation}}\right)-1\right)}}.$$

($\cancel{4}$5)

$\cancel{\text{The heat map in Fig. S13 gives the result of this equation for different values of } k_R \text{ and } \tau_{\text{truncation}}.}$

[Figure]

[Figure]

**Figure S18:** The MNR-ratio of adiabatic and rectangular pulses as a function of the background density (with rectangular pulses) and the $\tau_{\text{truncation}}$ time. The corresponding maximum distance according to equation (13) of the main text is also depicted. As the background density reflects the concentration the x-axis is a measure for the concentration of the spin centres. In our sample with 80 µM, we had a background density $k_{\text{R}}$ of 0.1 with rectangular pulses.

~~Our results hint that the performance of shaped pulses can heavily depend on the circumstances of the measurement. For long traces and high concentrations, where a strong background decay has to be expected, broadband shaped pulses can further increase this decay and therefore increase the noise level. For short traces and low concentration on the other hand the increase in modulation depth due to broadband shaped pulses outperform the steeper background decay. Note that for our measurements, we typically had a $k_R$ of about 0.1 at a concentration of 80 µM with rectangular pulses. This means that a value of 0.2 would correspond to a concentration of 160 µM which is a lot larger than what is needed for most practical applications (Jeschke, 2012). This means that in all relevant cases a sensitivity increase can be expected when broadband shaped pulses are used. Particularly, broadband shaped pulses perform better for lower concentrations and for shorter distances who allow to pick a shorter dipolar evolution time. For practical applications, a sensitivity increase due to adiabatic pulses is mostly desirable for samples with large distances and low concentrations who typically suffer the most from a low sensitivity. Our results hint that whereas for small concentrations a MNR improvement owing to broadband shaped pulses can be expected, however, this increase gets worse for longer dipolar evolution times.~~

Figure S18 shows that the performance of shaped pulses can heavily depend on the circumstances of the measurement. For a maximum distance below 4 nm ($\tau_{truncation} \approx 5$ µs), a MNR increase can be expected for all realistic concentration ranges. This is not the case if a longer distance shall be detected. For maximum distances around 5 nm, the MNR increase goes to 1 for high background densities of $k_R = 0.15$ 1/µs, which corresponds to very high concentrations > 100 µM. Typical concentrations for DEER measurements are around 50 µM, which here corresponds to a $k_R \approx 0.06$ 1/µs. For this concentration, a significant increase in the MNR can only be expected up to a truncation time of $\tau_{truncation} = 10$ µs, which is equal to a maximum distance of approximately 6 nm.

As broadband shaped pulses are particularly interesting for long distances, the calculations were performed up to a rather long truncation time of 25 µs (maximum distance of approximately 7.5 nm). For distances in the range > 6 nm, only with concentrations in the range of 10-30 µM ($k_R \approx 0.01$-$0.04$ 1/µs) a significant increase in the MNR due to broadband shaped pump pulses can be expected. The MNR increase drops quickly when higher concentrations are used. For a maximum distance of 7.5 nm and for concentrations over approximately 40 µM no increase can be expected any more due to broadband shaped pulses. If a concentration of 80 µM is used, the MNR is about to decrease to roughly 40 % when switching to broadband shaped pulses. It is known that diluting the sample is favourable if long distances shall be detected because it increases the phase memory time of the echo (Schmidt et al., 2016). When broadband shaped pump pulses, the higher background decay adds an additional point for carefully choosing the concentration of the sample and it seems to be advisable to avoid high concentrations.

**S22 Comparison of the resonator profiles**

Figure S19 shows the resonator profiles of the measurement of the sample with the high and the low concentration. The $B_1$ strengths that have been achieved for the sample with the low concentration were a bit lower.

[Figure]

**Figure S19:** Resonator profiles for samples with an 80 μM concentration (green) where all optimisation measurements have been performed and for the sample with a 30 μM concentration (blue).

**S21 Supporting Information References**

Doll, A. and Jeschke, G.: Fourier-transform electron spin resonance with bandwidth-compensated chirp pulses, J. Magn. Reson., 246, 18–26, doi:10.1016/j.jmr.2014.06.016, 2014.

Fábregas Ibáñez, L. and Jeschke, G.: Optimal background treatment in dipolar spectroscopy, Phys. Chem. Chem. Phys., 22(4), 1855–1868, doi:10.1039/C9CP06111H, 2020.

Jeschke, G.: DEER Distance Measurements on Proteins, Annu. Rev. Phys. Chem., 63(1), 419–446, doi:10.1146/annurev-physchem-032511-143716, 2012.

Jeschke, G., Chechik, V., Ionita, P., Godt, A., Zimmermann, H., Banham, J., Timmel, C. R., Hilger, D. and Jung, H.: DeerAnalysis2006—a comprehensive software package for analyzing pulsed ELDOR data, Appl. Magn. Reson., 30(3), 473–498, doi:10.1007/BF03166213, 2006.

Schmidt, T., Wälti, M. A., Baber, J. L., Hustedt, E. J. and Clore, G. M.: Long Distance Measurements up to 160 Å in the GroEL Tetradecamer Using Q-Band DEER EPR Spectroscopy, Angew. Chem. Int. Ed., 55(51), 15905–15909, doi:10.1002/anie.201609617, 2016.

Stoll, S. and Schweiger, A.: EasySpin, a comprehensive software package for spectral simulation and analysis in EPR, J. Magn. Reson., 178(1), 42–55, doi:10.1016/j.jmr.2005.08.013, 2006.

Tait, C. E. and Stoll, S.: Coherent pump pulses in Double Electron Electron Resonance spectroscopy, Phys. Chem. Chem. Phys., 18(27), 18470–18485, doi:10.1039/C6CP03555H, 2016.